# A sedimentary ancient DNA perspective on human and carnivore persistence through the Late Pleistocene in El Mirón Cave, Spain

Pere Gelabert [1,2,11] ✉, Victoria Oberreiter [1,2,11], Lawrence Guy Straus[3,4], Manuel Ramón González Morales [5], Susanna Sawyer[1,2], Ana B. Marín-Arroyo[4], Jeanne Marie Geiling [4], Florian Exler[1,2,6], Florian Brueck [1,7], Stefan Franz[1], Fernanda Tenorio Cano[1], Sophie Szedlacsek[1], Evelyn Zelger[1], Michelle Hämmerle [1,2], Brina Zagorc [1,2], Alejandro Llanos-Lizcano [1,2,8], Olivia Cheronet [1,2], José-Miguel Tejero [1,2,9] ✉, Thomas Rattei [10], Stephan M. Kraemer[2,6] & Ron Pinhasi [1,2] ✉

Caves are primary sites for studying human and animal subsistence patterns and genetic ancestry throughout the Palaeolithic. Iberia served as a critical human and animal refugium in Europe during the Last Glacial Maximum (LGM), 26.5 to 19 thousand years before the present (cal kya). Therefore, it is a key location for understanding human and animal population dynamics during this event. We recover and analyse sedimentary ancient DNA (sedaDNA) data from the lower archaeological stratigraphic sequence of El Mirón Cave (Cantabria, Spain), encompassing the (1) Late Mousterian period, associated with Neanderthals, and (2) the Gravettian (c. 31.5 cal kya), Solutrean (c. 24.5–22 cal kya), and Initial Magdalenian (d. 21–20.5 cal kya) periods, associated with anatomically modern humans. We identify 28 animal taxa including humans. Fifteen of these taxa had not been identified from the archaeozoological (i.e., faunal) record, including the presence of hyenas in the Magdalenian. Additionally, we provide phylogenetic analyses on 70 sedaDNA mtDNA genomes of fauna including the densest Iberian Pleistocene sampling of *C. lupus*. Finally, we recover three human mtDNA sequences from the Solutrean levels. These sequences, along with published data, suggest mtDNA haplogroup continuity in Iberia throughout the Solutrean/Last Glacial Maximum period.

During the Late Pleistocene (Marine Isotope Stages 5-2), Neanderthals and early anatomically modern humans (AMH) were competing with large carnivores for the occupation of the same ecological niches[1,2]. This competition included access to vital resources such as food and shelter[3–9]. Notably, during the Late Middle and Early Upper Palaeolithic, the primary carnivore contenders for these resources were the cave lion (*Panthera spelaea*), leopard (*Panthera pardus*), cave hyena (*Crocuta crocuta spelaea*), wolf (*Canis lupus*) and dhole (*Cuon alpinus*)[10]. Many archaeological cave sites with Middle and Upper Palaeolithic occupational phases exhibit compelling evidence of alternating occupations between carnivores and humans, especially when these occupations were seasonal[11,12]. These caves are thus invaluable repositories for insights into the adaptations and interactions of species during the Upper Palaeolithic (c. 45 to 12 thousand

years ago [kya])[13] while shedding light on the behaviour of AMH[3]. However, the physical presence of such animal evidence is very scarce, complicating the assessment of the co-occurrence of humans and particular animal taxa.

For the most part, archaeozoological studies rely on the morphological identification of animal remains and the taphonomic aspects affecting such remains (i.e. carnivore and/or anthropic and other biotic and abiotic agents of modification) to determine the presence of taxa in the available archaeological record[14]. These analyses have also shed light on the distribution of prey species, some potentially related to human subsistence and occupation[15,16]. Recently, palaeoproteomic methods, particularly Zooarchaeology by Mass Spectrometry (ZooMS)[17], enable the detection and charting of taxa from skeletal fragments lacking taxon-specific features[18–23]. However, ZooMS identification is restricted to preserved skeletal material and, as such, cannot reveal traces of vertebrate species that may have frequented the cave. Their remains are not necessarily recovered or part of the archaeozoological assemblage. Moreover, despite recent improvements, ZooMS only enables taxa identification without further phylogenetic inference and may lack resolution at the species level. Hence, the analyses of ancient DNA from sediment (sedaDNA) can provide additional insights by identifying the presence of animal species or human groups in archaeological sites and recovering their DNA even without visible skeletal remains[24–27].

Dated skeletal remains from Europe reveal a significant decline and eventual disappearance of multiple Pleistocene taxa at the end of the Late Pleistocene (Marine Isotope Stages 3 and 2, c. 60–10 kya). Cave hyenas (*Crocuta crocuta spelaea*) are believed to have disappeared from the European archaeological record around 31 kya[28], although there is potential evidence of their later persistence in southern Europe[29]. The faunal record shows cave lions (*Panthera spelaea*) persisted in Europe until 14 kya[30–34], potentially surviving longer in some areas. In northern Atlantic Iberia's Basque Country, the Armintxe cave paintings provide indirect evidence of lions' presence during the Magdalenian[35]. The leopard (*Panthera pardus*)[31–33,36,37] and dhole (*Cuon alpinus*)[10,14,38] eventually vanished at the end of the Pleistocene. Due to the limited amount of remains, the exact extirpation dates cannot be ascertained.

The Cantabrian region in Northern Spain was one of the main European human refugia during the Last Glacial Maximum (LGM, between 26.5 and 19 cal kya[39]) and has some of the best-preserved Upper Palaeolithic assemblages. Palaeoclimatic and palaeoenvironmental conditions during the LGM involved extreme temperature fluctuations and the expansion of ice sheets, which consequently limited the extent of human-inhabited areas across Europe[40,41]. Recent studies have revealed genomic homogeneity among European populations during the pre-LGM Gravettian period (33–24.5 kya)[42,43]. Later, part of this gene pool survived in the Franco-Cantabrian refugium with genetic continuity persisting during the Solutrean (24.5–21 kya)[44] and the succeeding post-LGM Magdalenian archaeological, cultural period (21–12 kya)[43,45].

Here, we present our results for the study of sedaDNA data from the lower archaeological stratigraphic sequence of the El Mirón Cave (Cantabria, Spain) vestibule rear, north section (Fig. 1, Supplementary Fig. 1, Supplementary Note 1). The site has a culture-stratigraphic sequence of levels dated by 101 radiocarbon determinations from >46,000 to c. 4000 cal BP[46,47]. Remarkably, it exhibits excellent organic preservation, the faunal record has been studied and published and was continuously occupied through the LGM, making it an ideal site to study the genomics of humans and fauna through this period. The results show 1) the identification of mammalian species through sedaDNA, which are absent or rare in the studies of the archaeofaunal assemblages, including the late persistence of carnivores such as hyenas and leopards in Iberia, evidencing cooccurrence with humans, 2) the discovery of undocumented genetic phylogenies

through the study of 70 mtDNA sequences from 11 fauna species recovered from El Mirón sediments, and 3) human mtDNA continuity through the LGM, pointing towards genetic stability.

## Results

### Patterns of DNA presence in El Mirón

We screened 32 sediment samples (Supplementary Data 1, Supplementary Table 1) from the rear vestibule of El Mirón cave (deep *sondage* meter-squares V, W, X10) (Fig. 1, Supplementary Fig. 1, Supplementary Note 1) spanning from the late Mousterian to the Initial Magdalenian (>46,000–21,000 cal BP). We used a mitochondrial (mtDNA) in-solution capture designed at the University of Vienna and manufactured by Twist[48], including 51 mammalian species for retrieving sedaDNA from a broad spectrum of taxa. After a strict classification process (Methods, Supplementary Note 3), we detected the presence of 31 taxa of animals (including humans), 28 of which had congruent signals of aDNA based on length and deamination (Supplementary Data 2, 3, Supplementary Note 3). The number of recovered species and DNA amounts varies across the profile (Fig. 2). In the samples from this study, DNA preservation is related to archaeological level age, and lower levels show less preservation (Supplementary Fig. 4, Supplementary Data 4 and 5). Based on a PCR assay, we tested the inhibitory effect of the purified extracts, and we found that the lower DNA yields of level 130 are not related to inhibition and are likely due to poor organic preservation (Supplementary Note 2, Supplementary Figs. 3, 4, Supplementary Data 4 and 5, Supplementary Table 2).

The archaeological evidence of human activity in the Mousterian (Level 130), Early Upper Paleolithic (Levels 129-128) or even Solutrean (Levels 127-121) periods is far less abundant than in the Late Upper Palaeolithic (Magdalenian). These Levels (130-121) are located at the rear of the cave (squares V-W-X10, 2–3 m² of excavated area), (Fig. 1B, Supplementary Fig. 1): Level 130 yielded 115 lithic artefacts; in comparison, 573 lithic elements have been recovered from Level 128[49]. However, none of the levels 130, 129 and 128 yielded evidence of hearths or fire-cracked rocks that would have suggested intense human occupation in this part of the cave and during these periods[49,50]. The overlying Solutrean levels (127-121) show an increased density of archaeological remains and organic matter with seasonal, but somewhat more intense occupations[11,50]. In this study, we observe that the lowest level of El Mirón (130) presents the lowest amounts of aDNA (Supplementary Note 3) and also the lowest number of identified animal species through sedaDNA (Supplementary Data 3). While the average in our dataset is 10.6 species per sample, only one sample from level 130 yielded 10 identified species (Fig. 2, Supplementary Data 3). Overall, the distribution of prey taxa is in agreement with the archaeofaunal descriptions. However, this is not the case for carnivores and uncommon taxa (Fig. 2, Supplementary Note 3, Supplementary Data 3, 6 and 7). We did not recover endogenous human DNA from level 130, but traces of contamination (Supplementary Data 3 and 6, Supplementary Note 3).

### Detection of undocumented species at El Mirón through sedaDNA

Carnivores and some herbivores, such as reindeer (*Rangifer tarandus*), are present in relatively low numbers in Cantabrian archaeofaunal Upper Palaeolithic assemblages[11,44,50,51]. Many of the especially larger carnivores and bears are usually not observed during periods of human occupation, suggesting alternating use of certain caves by humans, carnivores, and bears[5,11,14,18,44,50–54]. We successfully recovered sedaDNA of multiple species that are not identified among the faunal remains, namely *Crocuta crocuta, Lynx pardinus, Cuon alpinus, Rangifer tarandus, Falco sp., Columba livia, Pyrrhocorax sp., Sorex araneus, Talpa europea, Mustela nivalis, Coelodonta antiquitatis* and *Strix sp.* (Supplementary Data 3 and 7, Fig. 2). Furthermore, we discovered species at certain levels with no existing records of their physical

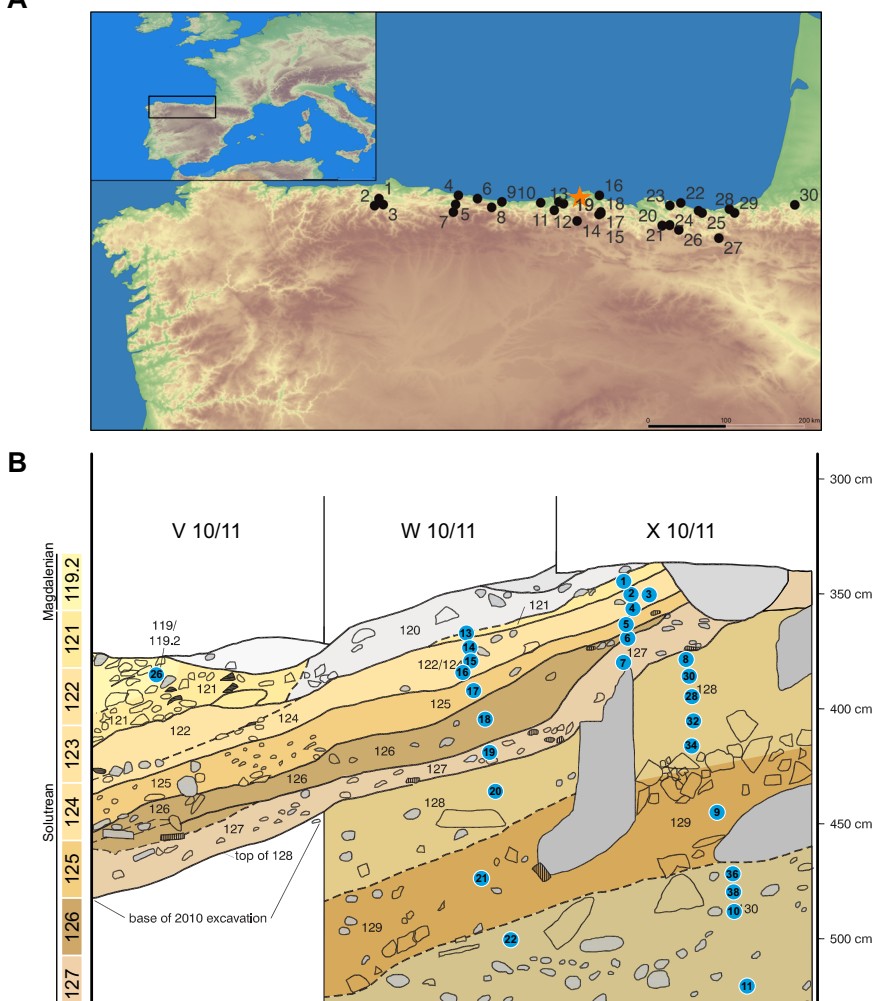

**Fig. 1 | Sediment samples from El Mirón. A** Location of El Mirón cave (denoted as an orange star) in the Cantabrian Region of northern Spain as well as some significant other caves located in this area: 1. La Paloma, 2. El Conde, 3. La Viña, 4. Tito Bustillo, 5. Los Azules, 6. La Riera, Cueto de la Mina 7. Collubil, 8. Coimbre, 9. Llonín, 10. Altamira, 11. Hornos de la Peña, El Castillo, 12. El Pendo, 13. Morín, 14. Rascaño, 15. La Garma, 16. La Fragua, 17. El Otero, 18. El Valle, 19. El Mirón, 20. Axlor, 21. Bolinkoba, 22. Cueva de Santa Catalina, 23. Santimamiñe, 24. Ekain, 25. Erralla, 26. Lezetxiki, 27. Coscobilo, 28. Ametzagaina, 29. Aitzbitarte III & IV, 30. Isturitz (French Basque Country). **B** El Mirón Cave NE corner of the vestibule rear stratigraphic section with locations of the collected samples (blue dots) in the north face of the V-W-X/10 excavation units, including the W-X10 deep *sondage*, numbers inside dots are the sample numbers (modified from L.G. Straus & R. L. Stauber).

remains in those levels, including those of *Panthera pardus, Mammuthus primigenius* and *Vulpes vulpes*. For example, archaeozoological remains of *Panthera pardus* have only been recovered from level 128, while we have genetic evidence of its presence in all the sampled levels except 122 (Supplementary Data 3). In addition, we also identified reindeer DNA in the undated and artefact-poor level 129 and the Gravettian-age level 128. Overall, its presence in both the sedaDNA and archaeofaunal registers is limited. Previously, it was only identified in the fossil record of El Mirón in the form of a grooved incisor pendant in the Lower Magdalenian level 17 (Cabin area, Supplementary Fig. 1)[44]. However, although not abundant, its presence has been identified in several archaeological sites, mainly in the Basque region[55] but as far west as Asturias, dating from the Aurignacian to the Magdalenian[18,56,57].

We have identified an average of 3.4 species of carnivores per sample, with eight samples from levels 129 to 119.2 showing up to 5 different species in a single sample (Supplementary Data 3 and Supplementary Table 1). Previous archaeozoological research on El Mirón levels 130-119.2 identified *Canis lupus* fragments only in levels 128 and 130, and no *Cuon alpinus* fragments have been reported[50]. In contrast, here we identify the presence of both canids (dhole and wolf) in all the studied levels of El Mirón (Supplementary Data 3 and 7, Fig. 2). Finally, based on the study of sedaDNA, we determined that while no cave bear (*Ursus spelaeus*) is present in El Mirón, genetic traces of brown bears (*Ursus arctos*) are present throughout the lower profile (Mousterian-Initial Magdalenian) of the site sequence.

In summary, we expand the temporal range of several carnivore species to later strata than previously documented. Specifically, we

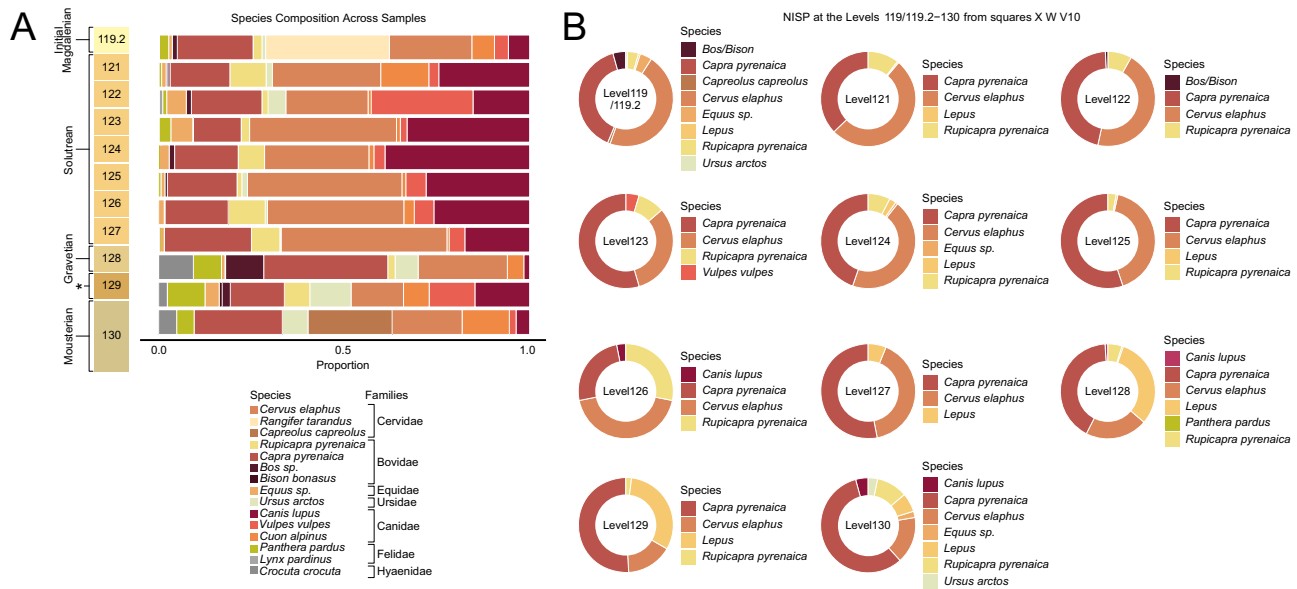

**Fig. 2 | Faunal distribution at El Mirón. A** Principal species identified with sedaDNA per archaeological level. The families with corresponding genera/species are displayed in the legend according to highest to lowest overall read presence and grouped by families. The X-axis displays the normalised read counts for each genus/species within each archaeological level, while the Y-axis shows the archaeological levels and associated human archaeological cultures. The asterisk denotes a level (129) with little human presence and undefinable archaeological culture. **B** NISP according to level (Supplementary Data 7).

push back the extinction timeline of spotted hyenas in Iberia to at least the Magdalenian/post-LGM period. Additionally, despite their limited representation or absence in the archaeological record, we provide evidence for the continued presence of species such as dhole and leopard in El Mirón through the MIS2 (Fig. 2, Supplementary Note 3).

## sedaDNA reveals multiple faunal mtDNA lineages at El Mirón

Our previous results focused on identifying the presence of mammalian mtDNA based on a classification of sedaDNA sequences. However, identifying the presence of specific species does not shed light on their genetic phylogenies. The remarkable DNA preservation at El Mirón enabled the assembly of 70 partial and complete mtDNA genomes to go beyond species identification and explore the phylogenetic relationships of 12 of the 28 identified species, including humans.

For the phylogenetic analyses, we focused on the samples with average coverage depths greater than 5X and signs of deamination (more than 0.4 at read end) (Methods, Supplementary Data 3, Supplementary Notes 2–4), which allowed us to assemble mtDNA genomes of 21 *Cervus elaphus*, 17 *Capra pyrenaica*, 14 *Canis lupus*, 6 *Rupicapra pyrenaica*, 4 *Vulpes vulpes*, 3 *Cuon alpinus*, 2 *Panthera pardus*, 1 *Rangifer tarandus*, 1 *Ursus arctos*, and 1 *Equus* sp samples. (Supplementary Data 8). These numbers indicate the individual samples for each species that had a coverage depth greater than 5X. All these genomes have high deamination patterns and missing sites (Supplementary Data 3). The missing sites are likely due to the stringency of the classification that has reduced the coverage across mammal-conserved mtDNA regions (Supplementary Data 3). We also observed variability in the mtDNA sequences in each sample, both using all the substitutions or restricting it to variable transversions to minimise the effect of damage (Supplementary Data 9). This suggests that each genome originated from multiple individuals, a scenario that precludes the feasibility of conducting calibrated phylogenies for estimating split times.

Only seven bone fragments of *Canis lupus* have been identified, where sedaDNA levels have been studied (Supplementary Data 7 and Supplementary Table 1). In contrast, all the levels except 119.2 and 130 yielded enough mtDNA to reconstruct partial *C. lupus* mitogenomes with average coverages ranging from 5.39X to 58.51X, These partial mitogenomes represent the densest up-to-date sequencing of mtDNA

Pleistocene *C. lupus* of Iberia. The mitogenomes were aligned with Palaeolithic and modern *C. lupus* mitogenomes[58–61]. First, we observe that wolf lineages are diverse in time and space, all showing phylogenetic relationships with Pleistocene wolves of Europe[58,60,62]. Some El Mirón sequences cluster close to Palaeolithic wolves from Goyet Cave, Belgium[61] (Fig. 3, Supplementary Fig. 11), showing temporal similarity. The oldest available mtDNA sequences of Pleistocene European dogs fall close to or within dog clades A, C and D[59,60,63,64]. All the newly reported sequences from El Mirón fall out of this diversity and are close to the oldest wolves of the dataset (Fig. 3). Therefore, we can only confirm the presence of wolves in the Solutrean sequence and not the presence of domestic *C. lupus* lineages[60].

From levels 129, 128 and 127 we also recovered the partial genomes of dholes (Supplementary Fig. 7), with remains previously found in northern Atlantic Spain only in the Lower Magdalenian of nearby Rascaño[38] cave. The dhole mtDNA from El Mirón resembles one Pleistocene mtDNA sequence from Bacho Kiro (Bulgaria) and expands the support of the existence of at least two different haplotypes of *Cuon alpinus* in Europe[65] differentiated from the modern dholes (Supplementary Fig. 7). We further identified four *Vulpes vulpes* genomes from levels 129, 126, 122, and 121. The *V. vulpes* sedaDNA mtDNA genomes all fall within the range of *V. vulpes* modern diversity (Supplementary Fig. 14).

*P. pardus* is the only species from the genus *Panthera* identified at El Mirón through archaeological faunal analyses. Our sedaDNA analysis confirmed this result. We recovered two partial genomes of *P. pardus* from levels 129 (46890–33160 cal BP) and 125 (22980–22240 cal BP), showing nucleotide diversity that suggests multiple individual origins. We generated a Maximum Likelihood tree that shows that both El Mirón sequences form a clade closely related to the BAR001 genome from Mezmaiskaya Cave (Russia, Northern Caucasus)[66] (Fig. 4, Supplementary Fig. 19). Hence, the El Mirón mtDNA consensus sequences are more similar to 35.000 YBP sequences from the Caucasus than the Baumannshöhle (Germany, ~40 kya) genomes, meaning that the *P. pardus* from Europe would not be a monophyletic clade providing further complexity to the scenario suggested in Paijmans et al.[66].

We recovered *Ursus arctos* DNA from all periods. However, only the El Miron_9 sample from level 129 has enough DNA to recover a

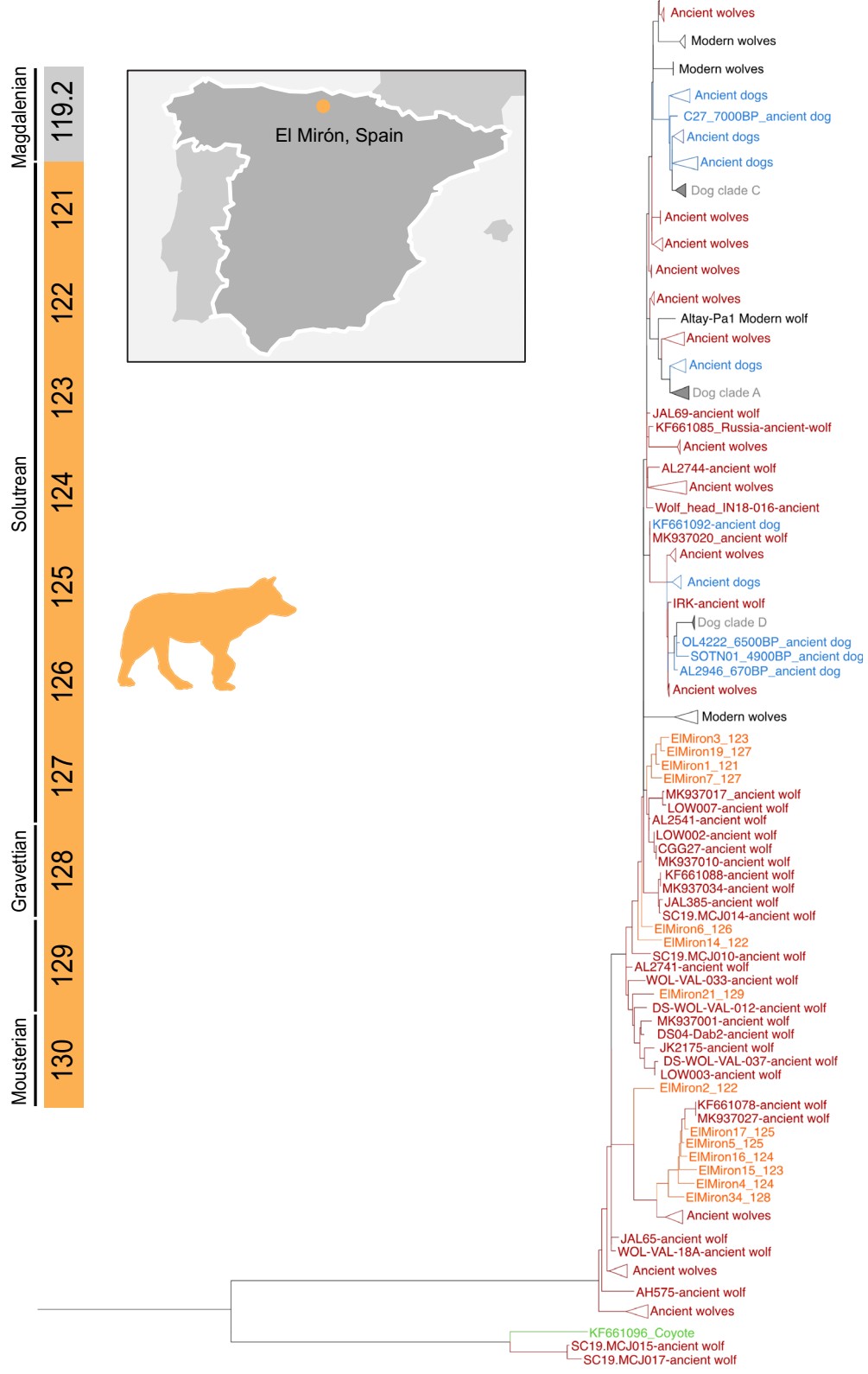

**Fig. 3 | Maximum likelihood tree of the *Canis lupus* mtDNA genomes.** Orange denotes the *Canis lupus* mtDNA genomes from El Mirón sediment samples. Ancient wolf mtDNA genomes from other locations are coloured in red. Modern wolves are coloured light grey, Modern dog clades are coloured grey, and Ancient dogs are coloured blue. The green colour is used for a Coyote genome. The tree is rooted with dhole mtDNA sequences, including two from El Mirón (Supplementary Fig. 11).

10X mtDNA genome. The *U. arctos* genome from El Mirón belongs to Clade 3a with other European genomes but is not similar to other Pleistocene genomes from Spain, which are present in Clade 1[67]. These published genomes are, however, much younger and could

indicate recent changes in mtDNA diversity in Iberia (Fig. 5A, Supplementary Fig. 22).

One species, *Crocuta crocuta spelaea*, did not have sufficient DNA preservation to produce mtDNA genomes using our 5X coverage

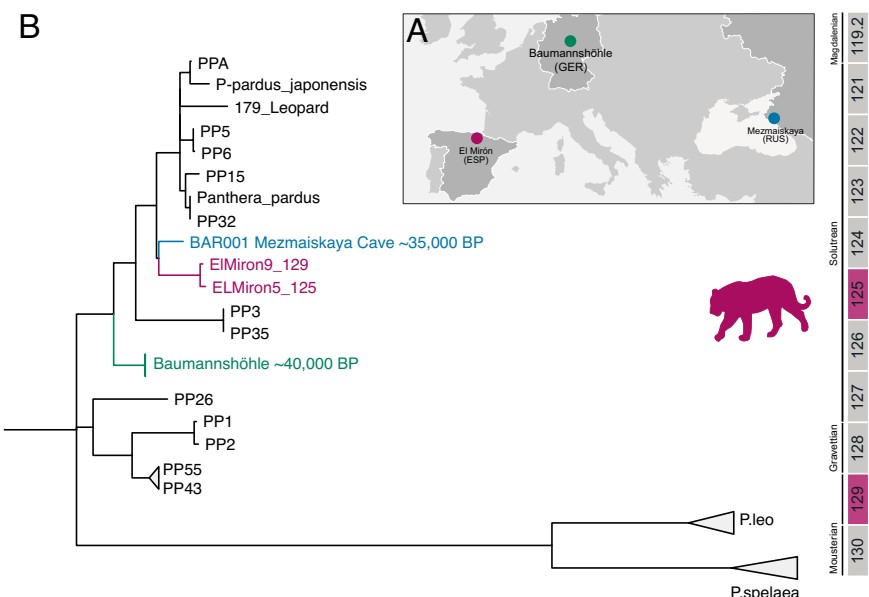

**Fig. 4 | Maximum likelihood tree of *P. pardus*. A** Location of the Pleistocene available mtDNA sequences, **B** genomic similarity between El Mirón leopards and Caucasus *P. pardus* BAR001 from the Pleistocene is observed. Circles represent the *P. pardus* mitogenomes: El Mirón (purple), Baumannshöhle (green), and Mezmaiskaya (blue).

filtering. However, we used the reads from ElMiron_10 from level 130 to assess its phylogenetic connection. The most recent fossils of *Crocuta crocuta spelaea* in Iberia are dated to 23.9 kya[68], with a single dated coprolite attributed to hyena (12,780 cal BP), suggesting potential persistence in Iberia until the late-glacial period[29]. The currently available data from Palaeolithic hyenas suggest diversity in mitochondrial haplotypes in Europe[69]. We thus performed a distance matrix analysis with the available mtDNA genomes from ancient and present-day *C. crocuta* (Supplementary Data 10) to understand the genetic relationship of the ElMiron_10 with other hyena populations. The Hyena mtDNA reads from level 130 of El Mirón and are most similar to *C. crocuta* specimens from Haplogroup A[69]. Haplogroup A is the same as CC8 and CC9 from France, dated to 22.6 kya[70].

Finally, we recovered partial mtDNA genomes of multiple herbivores. *Cervus elaphus* mtDNA genomes with coverages ranging from 5.5X to 63.78X could be reconstructed from levels 129 to 121. (Supplementary Data 3 and 8). Phylogenetically, they form a clade with the Solutrean-age genome from another Palaeolithic cave site (Cueva Chufín) in western Cantabria[48] and other sequences from Denmark and Poland. A Maximum Likelihood tree (Fig. 5B, Supplementary Fig. 26) shows that all the sequences from El Mirón are placed in Western Clade A[71] and closely related to the previously reported Iberian Pleistocene *C. elaphus* with the exclusion of some Late Pleistocene *C. elaphus* from Northern Spain in the palaeontological site of Liñares cave[72] forms a separate clade. Five partial genomes of Spanish chamois (*R. rupicapra pyrenaica*) and 17 partial genomes of Spanish Ibex (*Capra pyrenaica)* were recovered. The sequences of both species from El Mirón sedaDNA samples show little population diversity at the mtDNA level (Supplementary Fig. 34), and the sequences are congruent with those originating from multiple individuals. An mtDNA genome of *Rangifer tarandus* from level 129 falls close to *R. tarandus* sequences (Supplementary Fig. 26). Lastly, *Equus* sp. from the El Mirón Solutrean level 125 mtDNA is close to Pleistocene sequences and falls as an outgroup of clade A, B, and D[73] (Supplementary Fig. 37).

**Human mtDNA at El Mirón resembles Solutrean Iberian lineages**
We recovered and validated human DNA from 10 sediment samples. Every library produced from these sediment samples shows the typical deamination pattern and congruent read length distribution (Supplementary Data 6). The number of recovered filtered reads varies, and

only three samples from Solutrean levels presented enough reads to perform analyses: Sample ElMiron_1 from level 121 (2.5X), sample ElMiron_14 from level 122 (5.6X), and sample ElMiron_18 from level 126 (6.4X) (Supplementary Figs. 39–42). The contamination estimates of Schmutzi are congruent with limited modern DNA contamination (Supplementary Data 6). Calico shows that the three sequences originate from multiple donors (Supplementary Data 6). The genotypes of all the relevant alleles are presented in Supplementary Data 11, suggesting multiple donor individuals but no modern contamination. Next, we called the majority sequence of these individuals using Schmutzi. The consensus sequences were analysed with Haplogrep 3 to determine the haplogroup of most calls. We successfully assigned two samples (levels 122 and 126) to haplogroup U2'3'4'7'8'9, and a sample from level 121 could not be assigned further than being identified as falling within R or U haplogroup diversity, typical for the Upper Pleistocene in Europe[43,74]. The sequences from levels 122 and 126 exhibit the same haplogroup as the Solutrean-age Malalmuerzo individual (~23,000 cal yr BP) in Andalucia (southern Spain)[75], and the Solutrean individual from La Riera (21,011-20,725 yr calBP)[43], located ca. 125 km west of El Mirón in Asturias. Both Malalmuerzo and La Riera individuals belong to the same genomic cluster described as the Fournol-cluster. The Fournol-ancestry refers to the Gravettian-like ancestry defined by an individual from Southern France that is thought to represent genetically the Iberian LGM genetic diversity[43]. We produced a Maximum Likelihood phylogeny with multiple modern and Pleistocene mtDNA sequences from Europe that evidence the connections between El Mirón samples and other Solutrean individuals (Fig. 6, Supplementary Data 12). So far the Haplogroup U2'3'4'7'8'9 has been identified in thirteen individuals[42,43,75]. All these individuals belong to the Gravettian, Solutrean or Magdalenian cultures, distributed from Poland to the Iberian Peninsula restricting this mtDNA diversity to individuals from 27,000 to around 13,000 years ago.

## Discussion
In this study, we present a high-resolution sedaDNA analysis of faunal mitochondrial genomes through the Ice Age in a refugium. This pioneering approach is combined with extensive previous archaeological and archaeozoological research conducted at El Mirón Cave[11,47,76–79]. Our findings do not contradict the archaeological evidence indicating relatively limited human activity and significant carnivore presence

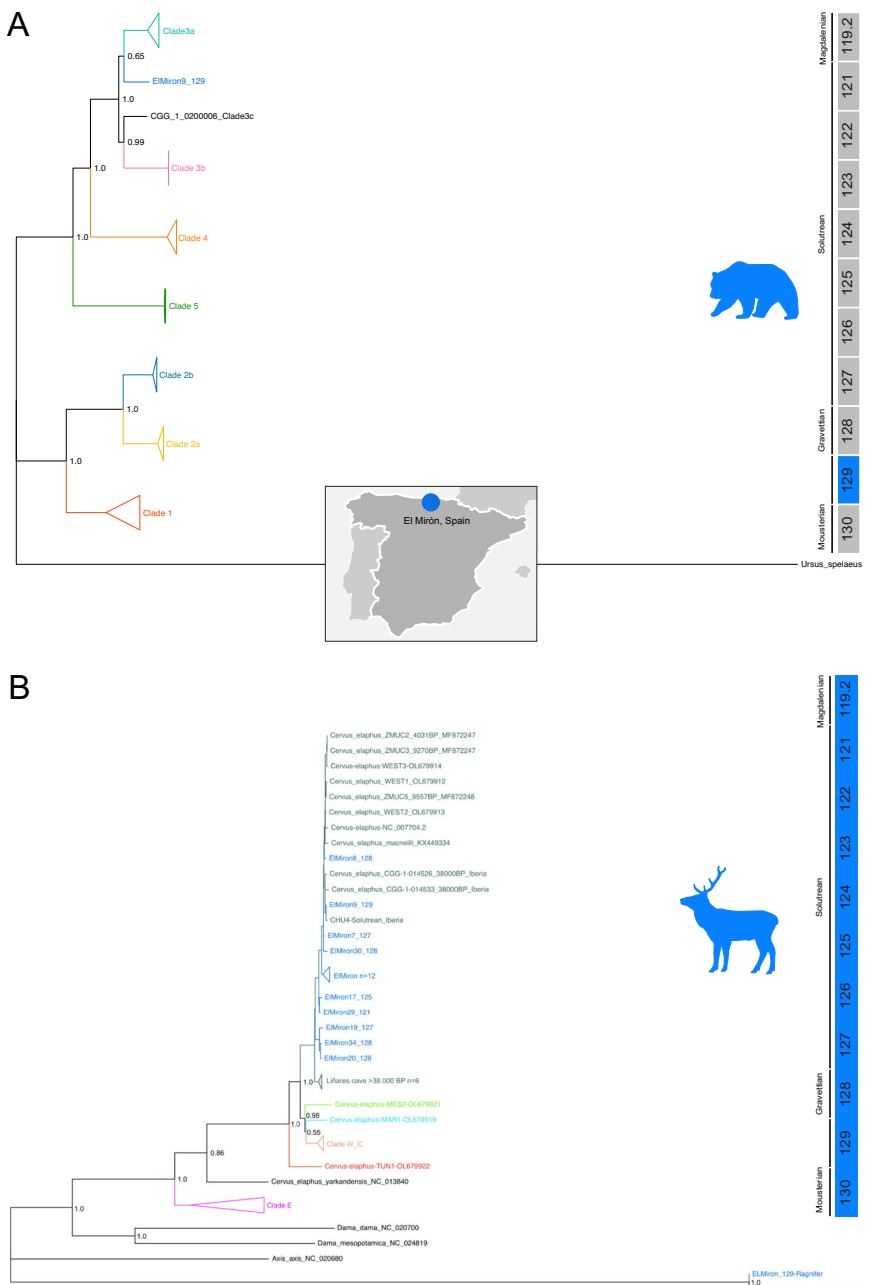

**Fig. 5 | Maximum likelihood tree of Brown Bear and Red Deer. A** A Maximum likelihood tree performed with 100 bootstrap replications suggests that the sample from El Mirón belongs to Clade 3a (**B**) A Maximum likelihood tree performed with 100 bootstrap replications places all the *Cervus elpahus* mtDNA sequences from El Miron in clade A (dark green).

during the Middle to Upper Palaeolithic transition in the rear vestibule of El Mirón Cave[50]. The sedaDNA analyses identified 28 taxa (6 carnivores and 21 herbivores and humans). In contrast, the faunal analyses identified only 15 species, highlighting the capacity of sedaDNA to detect the presence of uncommon taxa throughout the archaeological profile. We detected genetic material indicating that uncommon taxa such as hyenas and leopards survived in Iberia until the end of the Solutrean or even into the Initial Magdalenian periods. Additionally, we revealed the extensive presence of dholes in the Pleistocene, currently rare in Pleistocene assemblages[63]. Overall, we provide key genetic data that extends the known occurrence dates of Pleistocene megafauna in Iberia. Future research combining sedaDNA with other molecular methods, such as palaeoproteomics, may help identify rare, taxonomically undiagnostic fragments of these species[18,19], shedding light on their temporal dynamics.

We report the presence of carnivores (*C. crocuta, L. pardinus, P. pardus, C. lupus, C. alpinus* and *V. vulpes*) across all stratigraphic levels, with no evident differences in the number of carnivore species per sample except for notably lower preservation in level 130 and 119.2. Notably, levels 128–130 are situated near the cave walls in the rear part of the vestibule, an area closer to the dark inner cave. This location likely attracted carnivores more frequently than humans, as indicated by the limited archaeological evidence of human occupation in this part of the cave as observed in Amalda cave[7]. In the future, denser sampling in space and time could provide substantial data to identify quantitative differences and resolve occupation hiatuses, and date alteration patterns. We observed that older samples exhibit reduced DNA preservation, younger samples however show more inhibition which indicates an overall better preservation of organic material (DNA and inhibitors). Additionally, our data suggest that the

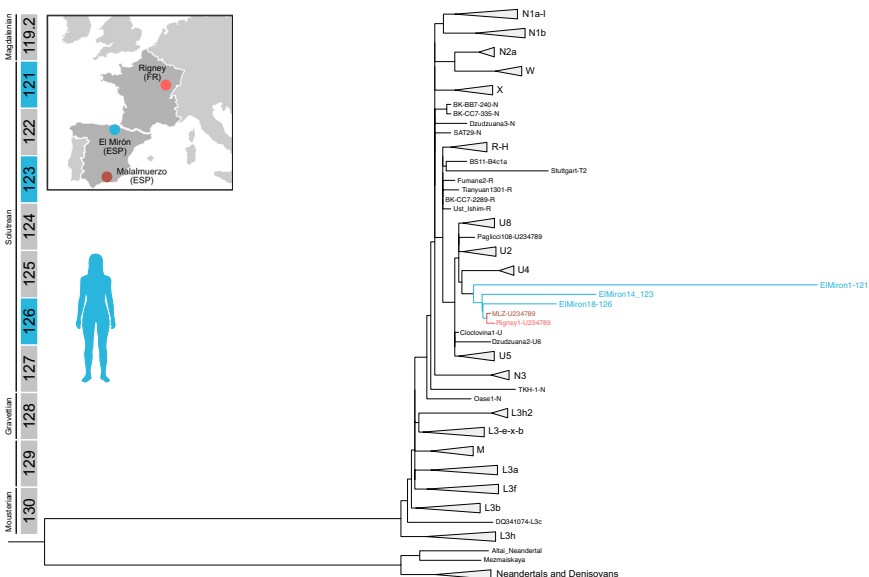

**Fig. 6 | Maximum likelihood tree of *Homo sapiens* performed with 100 bootstrap replications.** We observe that the three mtDNA sequences from El Mirón sediment samples from levels 121,122, and 126 (in blue) are located together and close to the individual from la Cueva de Malalmuerzo (brown)[75] and another individual from Rigney (France) (pink)[43]. El Mirón samples have long branches, suggesting an excess of substitutions added, a product of originating from multiple individuals.

archaeofaunal representation aligns more closely with sedaDNA findings for prey animals rather than carnivores. The persistent presence of carnivore DNA, coupled with their absence in the physical remains, indicates that carnivores frequently utilized the cave during the Palaeolithic. They likely scavenged leftovers from humans and intermittently occupied the cave in the absence of humans[14]. While carnivores may leave biological traces such as faeces and urine, their physical remains, such as teeth or bones, are less commonly found unless they perish in the cave. Future research linking sedaDNA to tissue origin might shed critical light on cave occupation patterns.

Beyond the ability to prove the presence of animals without any previous record in the cave, sedaDNA studies allowed the study of the genetic variability of the recovered animals, as previously demonstrated[24,26,80]. Specifically, sedaDNA enables two unique types of analyses that are not (or only very rarely) possible through the morphological study of remains: 1) The determination of the phenotypic/genotypic affinities of the data in comparison with other previously published specimens, 2) The identification of population or species replacements that are not visible in the osseous remains. In this work, we extended the genetic datasets of key species such as wolves, leopards, brown bears, and hyenas while describing previously undocumented differences in their genetic ancestry. In this direction, we detected at least two differentiated lineages of wolves living in the same area and a surprising mtDNA similarity of the leopards from El Mirón to the one from Mezmaiskaya Cave in the Northern Caucasus, proving that the mtDNA from leopards in Europe do not form a monophyletic clade. Finally, we observe that the hyena mtDNA from El Mirón shows similarities with Pleistocene hyenas from present-day France[70], suggesting regional connections at the mtDNA level at the Pleistocene.

In our study, sedaDNA was also used to provide evidence of the genetic stability of human mtDNA Solutrean lineages in Iberia. The human mtDNA data from three Solutrean levels at El Mirón show little diversity, all similar to two recently published Iberian Solutrean fossils[43,75], suggesting a stable refugium population in this region during and immediately after the LGM[75]. In the future, nuclear genetic data and denser sampling can date the population admixture that resulted in the Lower Magdalenian genetic pool represented by the Red Lady (El Mirón Ancestry)[42].

We have identified several limitations in our study that could impact our findings. Firstly, our focus area within the vestibule of El Mirón is limited to 2–3 m², whereas the entire vestibule site spans over 300 m². This specific focus, aligned with previous studies on the lower sequence of El Mirón may bias our results towards this particular area and its materials. This limitation suggests that different regions within the vestibule could yield different outcomes. Moreover, the location of our study area at the rear of the vestibule might influence the higher presence of carnivore DNA. A larger cave sampling may produce more robust statistical data, especially in the front vestibule[51,81]. In addition, future research microcontextual studies should be carried out to integrate the current results with the microstratigraphic context of the sediments at El Mirón[82], delving into the specific origin of the DNA and resolving possible secondary alterations that have not been identified so far with the current sedimentological, taphonomical, geological and chronological research at El Mirón[46,50,51,83–85]. Secondly, linking the amount of DNA discovered to specific activities or animal presence is challenging due to uncertainties regarding the tissue or organic material origins (i.e., faeces, saliva, urine, or other fluids). Our data proves the presence of a notable amount of DNA from species without remains. Further studies on DNA tissue origin can link animal activities and DNA presence. Third, we stretched the capacities to study ancient genomes due to the small amount of available mtDNA Pleistocene genomes of relevant species such as dholes or leopards. This limits our analyses and the inferences we can make. Finally, a potential limitation lies in the risk of DNA not contemporaneously deposited with the sediment. However, our rigorous validation methods and comprehensive analyses are designed to address and minimise these concerns effectively. We have evidenced the capacity to exclude the possible contaminant taxa and individually verify the identifications. The genetic similarity of our data to other Pleistocene sequences and the temporal dynamics strongly support the fact that the DNA is contemporaneous with the stratigraphic levels and available dates.

Our study provides an approach to understanding the complex relationships between archaeological and sedimentary aDNA data. Future research on DNA origins in sediments can complement these findings and archeozoological data. Pioneering studies have shown that most aDNA from sediments likely originates from faeces and skeletal elements[86]. This data can be cross-referenced with

archaeological findings and taphonomic indicators to determine whether the occurrence of certain species needs to be re-evaluated or if it indicates their absence in particular archaeological contexts. It can be regarded as a preliminary step towards new strategies for assessing faunal turnovers, identifying domestic forms of animals through sedaDNA, determining the extirpation dates of animal populations, analysing human activity patterns, and understanding the genetic affinities of both animal and human populations.

## Methods

### Sampling

Sampling was performed in the profile of the archaeological excavation trench at the rear of the El Mirón vestibule[46] in 2022 and 2023 (Fig. 1, Supplementary Data 1, Supplementary Fig. 2)[46]. After assessing the cave's stratigraphy, samples were taken with documentation of the locations of the excavation's 3D grid system and the archaeological context. 100 mg of sediment was taken for each sample, with control of contamination by using gloves and washing the sampling instruments with 5% bleach solution. Sediment was stored in sterilised bags and preserved in cold conditions until transported to the University of Vienna (Austria). The excavation team of the El Mirón Project has granted access and provided the necessary permissions to proceed with the current study. The use of sediments for destructive analyses addresses ethical concerns by using the little material as possible

### Experimental procedures

All samples were processed at the Palaeogenomics laboratory of the University of Vienna. We applied contamination control measures to mitigate the effect of modern DNA contamination. The samples were prepared in dedicated clean room facilities. We included negative controls at each step of the wet lab pipeline to control for potential contamination of reagents. 50 mg of sediment was extracted per sample following the protocol from Dabney[87] with adaptations from Korlević[88]. Sample El Miron_18 was extracted twice from two different 50 mg aliquots to increase the chances of retrieving human DNA. The DNA was eluted it in 50 μL TET buffer (10 mM Tris-HCl, 1 mM EDTA, 0.05% Tween 20, pH 8.0) and double-stranded libraries of 25 μL of the extract, as described in Meyer and Kircher[89], were prepared without shearing the DNA into smaller fragments. We used a MinElute PCR Purification kit from Qiagen to clean up the samples instead of SPRI beads and eluted in 40 μL EBT buffer (1 mM EDTA, 0.05% Tween-20). Positive control was added to each library batch using 24 μL of deionised water and 1 μL of a 1:250 dilution of CL104.

The number of PCR cycles per sample was determined through real-time PCR. Libraries were double-indexed[90] and amplified with PfuTurbo Cx HotStart DNA Polymerase from Agilent. A subsequent clean-up was performed with 1.2x NGS clean-up magnetic beads per sample, introducing a size selection. We eluted in 25 μL EBT buffer (1 mM EDTA, 0.05% Tween-20). Each indexed library was amplified further in preparation for solution capture using KAPA HiFi HotStart DNA Polymerase and IS5/IS6 as primer pairs[89]. We enriched the amplified libraries employing a custom-designed mitochondrial capture including human and 50 mammalian mtDNA sequences[48], following the TWIST capture protocol[91]. The capture was designed by TWIST biosciences as custom product with probes of 80 bp covering the whole sequences of the mtDNA of the 51 species (Supplementary Data 13). Following 16 h of hybridisation at 65 °C and four rounds of washing, we mobilised the target DNA from the probes in a PCR cycler at 95 °C for 5 min. Another qPCR was performed before amplifying half of the captured library using KAPA HiFi HotStart DNA Polymerase, and the primer pairs IS5/IS6. We cleaned up with magnetic beads. Quality controls were performed by applying Qubit and Tapestation. The captured libraries were sequenced at the Vienna BioCenter Core Facilities (VBCF) on an Illumina NovaSeq 6000 platform.

### Bioinformatics

Sequencing reads were demultiplexed. We removed adapters and reads shorter 30 bases and poly A tails were removed using Cutadapt 4.2[92]. Filtered reads were processed with SGA 0.10.15[93] and FASTX-toolkit 0.0.14[94] to remove reads with low qualities (base quality lower than 30 in 25% or more of the read length) and exact duplicates (using SGA preprocess with–dust-threshold=1, followed by SGA index with -a ropebwt option, and SGA filter with–no-kmer-check option). Filtered reads were then classified at the family level using euka[95], with a minimum cutoff of 50 fragments per taxa, an entropy value of 1.17, and a minBins value of 6. We also used the general deamination values to set as priors of the euka classification, minimising the contamination. This was set to 0.2.

We then processed each of the individual families separately. Classified reads by euka were then converted to fasta reads using Samtools 1.3[96] and analysed with BLASTn 2.16.0[97] using the whole nt database from NCBI (2023-08-17) and -outfmt 6 option. Classified reads were then analysed with MEGAN 6.25.6[98], and reads were classified at the genus level using the LCA approach and the taxonomy file from NCBI, setting a minimum support of 100 reads to validate the presence, selecting the top 10% with a 90% of identity. In cases where the number of 100 was clear at the species level, we also classified the reads at the species level and identified the species. This is common in taxa where only one taxon per genus is present. Only reads that were classified at the selected level were kept.

The classified reads at species/genus level were then converted to fastq reads with Samtools 1.3[96] and aligned with BWA aln 0.7.17[99] disabling seeding, setting a gap open penalty of 2 and an edit distance of 0.01 using the mtDNA reference sequence of the identified hit, if we could not resolve the species, we selected the most likely species reference sequence from that genus according to the literature (Supplementary Note 4). Reads were later filtered by read quality >30, and duplicates were removed using Samtools and Picard tools 3.0.0[100]. We used MapDamage 2.0[101] and Qualimap 2.0[87] to assess the mapping statistics. We then defined an identification at the species level when 50 reads were found, and the deamination values were >20% on both ends. To perform the read length distribution, we employed samtools[102], and the results were later plotted with R 4.1.2[103] (Supplementary Data 5).

### Single Taxa analyses

After alignment, individual animal genomes were studied, and we determined the deamination values, average read length, and the number of recovered reads. We explored the diversity within the genomes by determining the number of variable positions (Supplementary Data 9). This was restricted to positions with coverage greater than 5 using samtools 1.9 mpileup and bcftools 1.19[104]. The analyses were performed in all the sites and only on the transversions, to exclude the effect of damage. To further minimise the effect of damage in such calculation, only positions with MAF > 0.21 were considered, excluding positions in which only a single read was carrying the derived position.

We determined the presence of partial genomes when the average coverage of a taxon was >5X. In these cases, we called a consensus sequence using ANGSD 0.941[105] (calling the majority allele and a minimum coverage of 5). We also studied the sequence visually using IGV 2.16.1[106] and through variant calling using bcftools 1.21[107] to observe the presence of discordant alignments. The figures were designed by the authors using silhouettes from https://www.phylopic.org/ under CC0 license.

### Reporting summary

Further information on research design is available in the Nature Portfolio Reporting Summary linked to this article.

## Data availability

All software packages used for analysis are publicly available and cited in the Methods section or in the Supplementary Information. Data required to generate all figures in the manuscript are available in Supplementary Data files. Source data for Figs. 3, 4 5 and 6 can be obtained through the deposited reads. The remaining sediments are stored at the University of Vienna and can be acced through a consent. The genomic data (fastq format) generated in this study have been deposited in the European Nucleotide Archive database under accession code PRJEB74514.

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

## Acknowledgements

The research was funded by the MINERVA Research Platform grant (R.P., S.M.K., T.R., V.O.) of the University of Vienna. A HEAS Seed Grant funded J.M.T., P.G., and O.C. P.G. is supported by INEAL through STMG grant CA19141-8d068698. Research of J.M.T. is supported by the program Ramón y Cajal of the Spanish MCIN/AEI (MCIN/AEI/10.13039/501100011033. Project Number RYC2021-033759-I) and the European Community (NextGenerationEU»/PRTR). Excavations in El Mirón Cave directed by LGS and MRGM between 1996 and 2013 and continuing since 2022 with the additional direction of Igor Gutiérrez Zugasti and David Cuenca Solana, are authorised and partially funded by the Gobierno de Cantabria, with additional funding from the US National Science Foundation, Fundación M. Botín, Leakey Foundation, Ministerio de Educación y Ciencia, National Geographic Society, University of New Mexico, UNM Foundation Fund for Stone Age Research (J. and R. Auel, principal donors) and material support from the Town of Ramales de la Victoria and IIIPC Universidad de Cantabria. The campaigns from 2021 to 2023 have been financed through a nominative subsidy from the Department of Culture, Tourism and Sports of the Government of Cantabria within the framework of the project "The Prehistory of the Asón Basin: The caves of El Mirón (Ramales de la Victoria) and La Chora (Voto) ABMA is supported by SUBSILIENCE project (H2020-ERC-2018.CoG N. 818299). During his MSc, ALL was funded by Colfuturo (Fundación para el futuro de Colombia). Thanks to Paul Knabl for illustrating Figs. 1, 2 and 4 and assisting with other figures. Open access funding provided by University of Vienna.

## Author contributions

P.G., J.M.T., R.P., L.G.S., and M.R.G.M. conceptualised the study. O.C., M.R.G.M., V.O., and P.G. carried out the fieldwork (sampling). R.P., S.M.K. and T.R. acquired funding, V.O., F.B., S.F., F.T.C., S.S.z., E.Z., B.Z., F.E., A.L.L., and S.S. performed the experiments. P.G., V.O., M.H., A.B.M.A., and J.M.G. analysed the data. P.G., V.O., J.M.T., L.G.S., S.S., A.B.M.A. and R.P. wrote the text with input from all collaborators. LGS wrote the supplementary description of the site, its stratigraphy, industries and faunas.

## Competing interests

The authors declare no competing interests.

## Additional information

[1]Department of Evolutionary Anthropology, University of Vienna, Vienna, Austria. [2]Human Evolution and Archeological Sciences (HEAS), University of Vienna, Vienna, Austria. [3]Department of Anthropology, University of New Mexico, Albuquerque, NM, USA. [4]Grupo I+D+i EvoAdapta, Departamento de Ciencias Históricas, Universidad de Cantabria, Santander, Spain. [5]Instituto Internacional de Investigaciones Prehistóricas de Cantabria (Universidad de Cantabria, Gobierno de Cantabria, Santander), Santander, Spain. [6]Department of Environmental Geosciences, Centre for Microbiology and Environmental Systems Science, University of Vienna, Vienna, Austria. [7]Department of Botany and Biodiversity Research, University of Vienna, Vienna, Austria. [8]Facultad de Química y Farmacia, Universidad del Atlántico, Barranquilla, Colombia. [9]Seminari d'Estudis i Recerques Prehistòriques (SERP), University of Barcelona, Barcelona, Spain. [10]Division of Computational Systems Biology, Centre for Microbiology and Environmental Systems Science, University of Vienna, Vienna, Austria. [11]These authors contributed equally: Pere Gelabert, Victoria Oberreiter. ✉e-mail: pere.gelabert@univie.ac.at; jmtejero@ub.edu; ron.pinhasi@univie.ac.at

