## [Peer Review file · Nature Communications]

A sedimentary ancient DNA perspective on human and carnivore persistence through the Late Pleistocene in El Mirón Cave, Spain

Corresponding Author: Dr Pere Gelabert

Version 0:

Reviewer comments:

Reviewer #1

(Remarks to the Author)

The article titled "A sedimentary ancient DNA perspective on human and carnivore persistence through the Last Glacial Maximum in El Mirón Cave, Spain" analyses various samples of DNA extracted from sediments of Miron Cave, showing interesting results regarding the presence of animal species never before identified in the cave through traditional zooarchaeological means. The manuscript also provides important data on the survival of some animal species, such as hyenas, until very advanced dates, which is one of the most intriguing findings of this study.

As a reviewer, I believe my role should focus, given the multidisciplinary nature of this study, on my specific area of expertise, which is zooarchaeological and taphonomic studies. My expertise does not extend to DNA studies, so I believe that aspect should be systematically reviewed by specialists in this field. Therefore, I will focus my comments to the authors on specific aspects of the archaeological work, and more particularly on the zooarchaeological issue.

Firstly, I'd like to say that I believe approaches like this are of great interest to archaeological work. In Zooarchaeology, we face the significant challenge that it's not always possible to taxonomically identify faunal remains. Often, we have to make inferences based on the size of the animal and the general characteristics of the assemblage. A clear case is the analysis of Bovidae in African sites. Thus, these types of approaches are very important and help us better understand the use of anthropogenic spaces and the paleoecology of archaeological sites. Therefore, I would like to start by congratulating the authors for their efforts and their interesting results.

However, I have some comments that I believe could help improve the content of the manuscript, not the formal study itself. I think where the manuscript falls short is in the presentation of the site, which is somewhat lacking. A publication in Nature Communications deserves attention to such formal matters, even if it requires repeating information published in previous works, especially if it's information that can be included in the Supplementary Information (SI). Below, I will explain what I mean.

Firstly, I'm not entirely convinced that the title of the article aligns well with its content. I understand the focus on the Last Glacial Maximum, but the study analyses archaeological levels from the Middle Palaeolithic onwards. I think the title should more generally encompass the different chronologies addressed and not solely focus on the last period.

The introduction seems generally correct to me. It is well-structured and presents the ideas that will be further discussed. However, I believe it's important to highlight, when discussing the site, that ALL the faunal information available comes solely from a 3 m² area of the site. If I understand correctly, both from the manuscript description and the Supplementary Information (SI), all DNA and zooarchaeological data from the site studies come from the excavation squares V, W, and X10. I think it's important to explicitly include this information in the manuscript. It's a small excavation area, and although it provides high resolution, it remains a very limited surface area.

In Figure 1A, I believe other sites in the area could have been included to provide a clearer context for the site. After all, it's just a grayscale map, and there are two Spanish cities that have little relevance to the study's theme. I encourage the authors to include a more elaborate physical map showing the location of Miron Cave and other caves in the area. Cities can be included as references, but I think the other option is more interesting. Furthermore, and I emphasize this point, I think efforts should be made to improve Figure 2B and combine it with Supplementary Figure 1 since a site plan is important for understanding the stratigraphy being presented. I reiterate, it's only 3 m², a small area, and it's important for the reader to be able to assess the results based on that information.

Moving on to the results, but continuing with the same issue, in line 159, the authors say, "level 130 ONLY yielded 111 lithic artefacts." 159 artifacts may seem few compared to other sites, but the problem is that the size of the excavated area is not

clear. Please remove the value judgment of "ONLY" and try to rephrase that sentence.

Lines 165-168. According to the authors, the Mousterian levels clearly have a lower density of archaeological remains than the Upper Palaeolithic levels. What explanation do the authors provide for this fact? Does level 130 have a greater sedimentary thickness than the rest, suggesting it was less utilized by human groups? Are there diagenetic issues that could explain a lower density of archaeological material in this level?

And with this, I come to the main concern I have with this study. The work is conducted through the analysis of sediment samples from different levels. However, the geological context of the analysed sequence is not provided at any point. Only a figure of a field drawing with different colours depicting a three-meter-long section is included, but what is the grain size of the levels? What does the sedimentological analysis reveal? How were these levels formed? Why is there such a pronounced slope? Is this slope diagenetic, or is it attributed to the original topography? As you can see, I have quite a few questions about the geology just by looking at the section shown, which doesn't provide any information beyond the layering of levels. The authors may refer to previous publications to defend this, but I think it's mandatory to include a section in the Supplementary Information (SI) that thoroughly develops the stratigraphy of the analysed sequence. How can we trust the DNA results without being sure that the stratigraphic sequence hasn't been altered by hydrological, gravitational, or other processes? To exemplify this, we could talk, for example, about the case of El Pendo cave. As the authors may know, it's an important cave with a very long sequence, but recent geoarchaeological studies have shown that the sequence is mostly in a secondary position. If this same study were done in El Pendo cave without considering this data, could we trust the results? Consider this question as the main reason behind my inquiry.

Similarly, I believe it would be interesting to include not only a drawing of the analysed section but also some photographs. Additionally, I'm sure the authors have photographs of the sediment sampling process, so it would be good to include some in the SI.

The authors have already published some works on this site, including Geiling's embargoed thesis. I think it would be good, in addition to the geology, to include a section in the SI that develops the archaeological part of the site in a slightly more detailed way. It doesn't need to be overly developed, but I think it would be good to have a paragraph for each level detailing the types of lithic industry and fauna present in each level, as well as the type of human and carnivore activity identified in the levels (are they campsites? Carnivore dens?). It would also be helpful to include tables with more explicit numbers of remains. For example, the Minimum Number of Individuals (MNI) of the fauna, I believe, is very important to include in this study. Only the NISP value has been shown in the supplementary tables, and I think it's a rather poor piece of data that doesn't do justice to the extensive work done with the DNA data. Please improve the archaeological part in the SI; it will be a way to highlight the significance of the site and help readers understand the importance of the data, avoiding the misconception that because some of this data is already published, it's not worth including here. Nature Communications is a multidisciplinary journal, and therefore it's important that the archaeological part is also introduced, as it will help understand how the site functions.

In conclusion, I believe this is a good study, but I think the manuscript and SI fail to address a basic aspect: the context of the site. We could debate whether archaeological data are necessary or not, but in the case of geological data, I believe they are **MANDATORY** in this study because without them, we cannot validate the genetic data.

Reviewer #2

(Remarks to the Author)

Review for "A sedimentary ancient DNA perspective on human and carnivore persistence through the Last Glacial Maximum in El Mirón Cave, Spain" from Pere Gelabert et al.,

Gelaber et al. used animal mitochondria targeted sequencing to retrieve aDNA from past animals from sediment samples collected from the lower archaeological stratigraphic layers of El Miron (Spain). The novel contribution of this manuscript is the reconstruction of animal occupation history at El Mirón Cave during the Pleistocene using ancient DNA retrieved from sediments. The authors recovered ancient DNA from 29 animal species, about half of which were not known from the skeletal record. The authors provide evidence that some species were present for a longer time than what is known from the skeletal record, including some that were believed to have gone extinct earlier. Mitochondrial genomes were reconstructed for several species, allowing the authors to place them on a phylogeny. The genetic relationship to species outside El Mirón Cave permit inferences about relatedness across Eurasia and broaden the scope of the manuscript. However, while the data generated is interesting, the study's framing makes it difficult for the reader to understand the main point of the manuscript, what the author intends to convey, and what we can truly learn from this new data. The paper appears to be a methodological study but also shows promise in investigating interactions between different species during the Upper Paleolithic in the region, animal occupation at El Mirón Cave, and human mtDNA continuity in the area. While many of these points are suggested by the data, none of them have been thoroughly investigated, and at times, evidence supporting claims is not clearly reported, leading to some interpretations of the results sounding naive and the overall study appearing somewhat superficial. Overall the work and the data generated are interesting but the paper still requires major revisions in order to show how relevant these data are in a methodological and biological context. I suggest to the author to reframe the entire paper as a methodological study on using sediment ancient DNA to investigate the faunal occupation of an archaeological site, using El Mirón Cave as a case study, and include the other biological insights as peripheral outcomes. Of course, this suggestion is optional, provided that the authors finds a better angle and produce a more robust paper.

General major comments

1. The lack of numbered lines makes it difficult for the reviewer to pinpoint specific sections.

2. Each paragraph of the introduction seems to address a completely different study, lacking clear continuity and making it less engaging.
3. There is too much discussion in the results section. I would suggest to the authors to either combine the Results / Discussion in one section or clearly report the results with some context in the results section, saving the detailed interpretation for the discussion section. Currently the discussion section is not worth reading as it merely summarizes what has already been discussed previously, for instance the first paragraph of the discussion is very similar to the first paragraph of the introduction.
4. One of the major issues with the paper is that the authors don't discuss any potential limitations of their approach. Inferences using ancient DNA from sediment should address potential of DNA translocation between layers. The authors should contextualize the results concerning the integrity of the stratigraphic layers and discuss how consistent the results are in comparison.
5. Another important issue, is that the authors seem to interpret the amount of sequencing reads as an indicator of specie abundance at the site. This would assume that all the animals that were present at the site left equivalent traces of DNA relative to the time they spent, which is not true. Megafauna body decomposition at the site is likely to leave more DNA than a few human individuals just visiting the sites. Also, layers can have different characteristics leading to different property of aDNA preservation. Even in the same layer, the preservation can be different. A layer more conducive for aDNA preservation will generate more aDNA than a layer that is less. One cannot use amount of aDNA preservation as indicator of species abundance.

Specific major comments

1. The results presented in Figure 2 should be adjusted for sequencing depth before making claims about differences in species abundance between layers, as this could contribute to differences in the number of reads obtained per species.
2. The authors claim the results presented in Figure 2 are congruent with the archaeological findings as the Mousterian horizon had the lowest average read number for all species compared to the upper layers (lines 165-167). However, the results are also congruent with the preservation of ancient DNA over time, as less DNA is expected to be preserved in deeper layers than upper layers. No estimates of inhibition are provided, which could also contribute to differences in ancient DNA preservation across layers. These two alternative explanations are congruent with the results and should be mentioned in the manuscript. The claim also appears to be contradicted by the results presented in Supplementary Figure 2, where the average read number is highest in Mousterian layers for humans.
3. The authors should discuss if the results obtained from euka reported in Supplementary Figures 2 and 3 can be used to reliably support some claims (lines 172, 174). The results obtained with euka for humans do not appear to hold up after further processing the assigned reads and then mapping to the reference genome. For example, 6932 reads are assigned by euka to humans in Mousterian layers (Supplementary Table 2), some of which are presumed to be ancient given the deamination values set as priors (Supplementary Section 1), but 0% appear to display signs of deamination after mapping to the reference genome (Supplementary Table 8).
4. The claim of an increased presence of carnivores such as *Panthera pardus* in layer 126, which could indicate less human and other animal presence linked to the coldest period of the LGM (lines 173-175) do not appear to be supported by the data. Layer 126 appears to be the peak of human presence across all layers and there is no increased presence of *Panthera* (Figure 2).
5. Results supporting the claim that "the presence of elevated carnivore sedaDNA can be used as indicative of low-intense human occupation" (lines 387-389) should be provided. It's difficult to assess exactly which results, and the number of observations, support this claim. This claim warrants explicit evidence, given that human and large carnivore competition is a main section of the introduction.
6. While the phylogenies of ancient mitochondrial genomes permit the authors to place their results from El Mirón in a broader geographic context, the discussion of the results is lacking with little inferences drawn or interpretations offered. For example, the phylogenetic results of *Panthera pardus* depict the genomes from El Mirón being more closely related to the genome from Mezmaiskaya Cave in Russia than the genome from nearby Baumannshöhle in Germany (Figure 4). The authors say this finding is surprising (line 422), but don't explain why or provide an interpretation of this result.
7. There is also little interpretation provided regarding the phylogenetic results of *Canis lupus*, which depict different mitochondrial lineages that alternate in time at El Mirón, some of which are more closely related to those from Goyet Cave in Belgium (Figure 3). Are these faunal turnover events mentioned in the Discussion sections?
8. The discussion of the human mitochondrial phylogeny results (Figure 5) requires more context to understand the significance of the authors' findings. The term "Fournol" is used (line 351, 353) without defining it or providing any context. I find it difficult to follow the meaning of the sentence "All these Fournol-ancestry individuals have haplogroup U2'3'4'7'8'9, restricted to Southern Europe and may represent an early population of humans with Fournol genetic ancestry in Western Europe" (lines 351-354). More explanation is needed to justify the claim that this haplogroup is restricted to individuals from Southern Europe and how it represents an early population of humans with Fournol genetic ancestry in Western Europe, perhaps in the Discussion section. The La Riera individual mentioned in the results section is not labeled in Figure 5.

Specific minor comments

1. Figure 1: The number of samples collected from the layers as depicted by dots in Figure 1 does not always match the number of samples reported per layer in Supplementary Table 1 (e.g. layers 126, 128, 129)
2. Figure 2: Figure legends says species are depicted, but three labels are at the genus level (*Rupicapra*, *Bos*, *Panthera*). Species names should be in italics.
3. Figure 3: Figure presents results from *C. lupus*, but the figure caption incorrectly references Supplementary Figure 10 which is a Caprinae phylogeny
4. Supplementary Figure 2: Figure legend overlaps some text of the figure
5. Supplementary Figures 2 and 3: It would be helpful if the colors of the family names matched in these figures so that the reader can more easily compare the results.
6. Supplementary Section 2: It's reported that mammoth (*Mammuthus primigenius*) and woolly rhino (*Coelodonta antiquitatis*) DNA was retrieved from layer 130, but no data is reported in Supplementary Table 2 supporting this claim.
7. Supplementary Tables: Sample "El Mirón_18" is listed twice with different results, but no explanation why.
8. Methods: Incomplete first sentence (line 457) after the "Sampling" section
9. Methods: No information provided about the reference genomes used for mapping with bwa
10. Methods: It's stated that "We determined the presence of partial genomes when the average coverage of a taxon was >5X" (lines 476-477). However, the human mitochondria genome sequence retrieved from El Mirón_1 only has a coverage of 2.5X and is included without explanation.
11. There are differences in terminology used throughout the manuscript (e.g. "levels" vs "layers"; "euka" vs. "EUKA"; "genomes" vs. "sequences"). Perhaps the authors could chose one term if they have the same meaning.

Reviewer #3

(Remarks to the Author)

This manuscript entitled "A sedimentary ancient DNA perspective on human and carnivore persistence through the Last Glacial Maximum in El Mirón Cave, Spain". It is a very interesting and relevant study that shows how useful it is to apply new analysis techniques to the study of archaeological sites, since through sedimentary ancient DNA the presence of species that had not been documented from a fossil perspective can be documented. . Just because of this type of evidence, I believe that this article represents a new advance in the discipline, and although we know this type of analysis exists in archaeology, it has rarely been able to be applied with the success that has been applied here. This study highlights the excellent results that the Miron team has been achieving in recent years. However, although the research results are important, there are still aspects that the authors could develop in this article.

On the one hand, the authors analyze the presence of carnivores and humans based on DNA data. The study reveals the new presence of various carnivores, but new data on herbivores has also been observed. Given that the authors discuss subsistence strategies in the Middle and Upper Paleolithic, it would have been interesting for the authors to develop in one or two paragraphs what implications the recovery of DNA from herbivores that had not previously been discovered at the fossil level has. Would it be possible for humans to transport meat or skin from animals that have not left bone remains? This would be very relevant data, since it would imply transport strategies for possible prey, in the form of fleshed meat or skin

Minor commentes

Fig 2 is not well understood, I believe that this figure could be complemented with a table that shows the NISP and MNI of the taxa found in the Miron sequence by levels. In this way the information would be complete and easily visible to the authors. .

Line 64. Authors can cite more appropriate references e.g. This quote from Lindly (1988), the work of Jones & Carvalho (2023)

Lindly, J., 1988. Hominid and Carnivore Activity at Middle and Upper Paleolithic Cave Sites in Eastern Spain. *Munibe Antropologia - Arkeologia* 40, pp. 45–70

Jones, E.L., Carvalho, M., 2023. Ecospaces of the middle to upper paleolithic transition: the archaeofaunal record of the iberian peninsula. *J. Hum. Evol.* 177, 103331 <https://doi.org/10.1016/j.jhevol.2023.103331>

Line 68 More appropriate than ref 2 that is cited, which is very generic, would be the ref by Pérez Ripoll et al (2009), which is more specific and specific to the Iberian Peninsula.

Pérez Ripoll, M., Morales Pérez, M., Sanchis Serra, A., Aura Tortosa, J.E. & Sarrión Montañana, I. (2010). Presence of the genus *Cuon* in upper Pleistocene and initial Holocene sites of the Iberian Peninsula: new remains identified in

archaeological contexts of the Mediterranean region. *Journal of Archaeological Science*, 37: 437-450

Line 65. Ref 5 is not appropriate for this context as it refers to an Olduvai Gorge context that is unrelated to the topic discussed in this introduction. More appropriate may be any reference that shows alternation between humans and carnivores in the occupation of the sites, for example the recent publication by Pinto et al (2024) or Amalda (Yravedra et al 2011), the quotes from Llonin (Sanchis et al 2019). and others
Pinto et al 2024. Alternation between humans and carnivores in the occupations of the Mousterian site of Sopena ~ rock-shelter (Asturias, Spain). *Quaternary Science Reviews* Volume 328, 15 March 2024, 108468

Yravedra, J., 2011. A taphonomic perspective on the origins of the faunal remains from Amalda Cave (Spain). *J. Taphonomy* 8 (4), 301–334.

Sanchis, A., Real, C., Sauqu´e, V., Nunez-Lahuerta, C., Eguez, N., Tormo, C., Perez Ripoll, M., Carrion Marco, Y., Duarte, E., Rasilla, M. de la, 2019. Neandertal and carnivore activities at Llonin cave, Asturias, northern iberian peninsula: faunal study of the mousterian levels (MIS 3). *Comptes Rendus Palevol* 18 (1), 113–141. <https://doi.org/10.1016/j.crpv.2018.06.001>.

Ref 31 Change Altuna 1971 to Altuna 1972

Line 109-112: The authors do not include citations referring to the presence of some large carnivores existing in the Cantabrian region. For the Magdalenian there are records of *Panthera pardus* in the sites of El Castillo, El Juyo (González Echegaray & Freeman, 2015). . For *Panthera leo* there is a citation in Abauntz (Jerjotoma-Ortín et al 2024), el Juyo (see González Echegaray & Freeman, 2015) or La Paloma (Castaños, 1980), Cuon in Rascaño 4 (Altuna 1981). Which complement some of the quotes referring to sites such as Urriaga. Mentioned in the text.

Jerjotoma-Ortín et al 2024 The Mark of the Beast: a bone assemblage assessment from the North of the Iberian Peninsula (MIS 3). *Journal of Archaeological Science: Reports* 54 (2024) 104409

Altuna, J. (1981). Restos óseos del yacimiento prehistórico de Rascaño (Santander). In (González Echegaray, F. & Barandiarán, I. eds.) *El Paleolítico Superior en la Cueva de Rascaño*. Santander, pp. 221-269

Gonzalez Echegaray & Freeman 2015. Excavando la cueva del Juyo un santuario de hace 14000 años . *Monografías del Museo Nacional y Centro de Investigación de Altamira*, n.º 25

Castaños, P. 1980 LA MACROFAUNA DE LA CUEVA DE LA PALOMA (PLEISTOCENO TERMINAL DE ASTURIAS). En Hoyos M. et al 1980. *La Cueva de la Paloma*. Ministerio de Cultura.

Line 143-145. The authors indicate that they have been able to identify 51 species and refer to the supplementary files. However, these 51 species are not clearly seen in these files. As a suggestion, it would be important for the authors to put in a table which species are recognized at the fossil level and which are identified at the DNA level. It can be indicated as a presence/absence table.

Line 180-182: The authors cite refs 26, 28, 45-47, however these references are not the most appropriate. Ref 26-28, 45 refer to the extinction of some carnivores. The ref. 46 refers to a paleontological site without human presence. There are multiple Cantabrian sites with alternating humans, carnivores and bears in the occupation of the sites. Labeko koba Amalda, Abauntz, Llonin, Esquilleu, Sopena, Ekain, la Viña, Castillo, etc.

Line 377: The authors refer to two zooarchaeological and taphonomic studies of the transition between Neanderthals and anatomically modern humans in the Cantabrian Sea in relation to the activity of humans and carnivores, but they do not cite other works developed by other authors for the same region that also are relevant in this relationship between humans and carnivores eg Yravedra (2011, 2013), Pinto et al (2024), Sanchis et al (2019)

Sanchis, A., Real, C., Sauqu´e, V., Nunez-Lahuerta, C., Eguez, N., Tormo, C., Perez Ripoll, M., Carrion Marco, Y., Duarte, E., Rasilla, M. de la, 2019. Neandertal and carnivore activities at Llonin cave, Asturias, northern iberian peninsula: faunal study of the mousterian levels (MIS 3). *Comptes Rendus Palevol* 18 (1), 113–141. <https://doi.org/10.1016/j.crpv.2018.06.001>

Yravedra, J., 2011. A taphonomic perspective on the origins of the faunal remains from Amalda Cave (Spain). *J. Taphonomy* 8 (4), 301–334.

Yravedra, J., 2013. New contributions on subsistence practices during the Middle–Upper Palaeolithic in northern Spain. In: Clark, J., Speth, J.D. (Eds.), *Zooarchaeology and Modern Human Origins: Human Hunting Behavior during the Later Pleistocene*. Springer, New York, pp. 77–95

Pinto et al 2024. Alternation between humans and carnivores in the occupations of the Mousterian site of Sopena ~ rock-shelter (Asturias, Spain). *Quaternary Science Reviews* Volume 328, 15

Reviewer #4

(Remarks to the Author)

I co-reviewed this manuscript with one of the reviewers who provided the listed reports. This is part of the Nature Communications initiative to facilitate training in peer review and to provide appropriate recognition for Early Career

Researchers who co-review manuscripts.

Version 1:

Reviewer comments:

Reviewer #1

(Remarks to the Author)

For my part, this new version of the manuscript is perfectly publishable. I consider it is a contribution of great value for research in the north of the Iberian Peninsula. And in this case, the new manuscript has greatly improved the issues that I raised, e.g. the poor contextualization of the site or other problems of archaeological nature. I can only congratulate the authors and I hope to see the paper published.

In any case, as I said on the previous occasion, my specialty is not DNA, so I consider that the review of the technical parts of the article should be assessed by specialists in the field.

I have noted minor issues related to the use of italics words in the bibliography. Please, review the bibliographic list in order to use italics when species are referenced, as in the case of *Cuon*, e.g.

Pérez Ripoll, M., Morales Pérez, J. V., Sanchis Serra, A., Aura Tortosa, J. E. & Montañana, I. S. Presence of the genus *Cuon* in upper Pleistocene and initial Holocene sites of the Iberian Peninsula: new remains identified in archaeological contexts of the Mediterranean region. *J. Archaeol. Sci.* 37, 437–450 (2010).

Reviewer #2

(Remarks to the Author)

We have reviewed the initial submission of the manuscript titled "A sedimentary ancient DNA perspective on human and carnivore persistence through the Late Pleistocene in El Mirón Cave, Spain" by Gelabert et al., and we are generally very pleased with the revised version. The updated Figure 2, which illustrates the proportions of reads assigned to taxa, the experiment addressing inhibition, the discussion of limitations, the enhanced clarity of the results, and the more cautious interpretation of findings, along with the extensive supplementary material, all significantly improve the manuscript. The article is now more engaging, and the narrative is better articulated.

However, we still have additional points that we believe the authors should address:

Major Points:

1. Line 118 – In the introduction, when El Mirón Cave is first mentioned, please include the region and country, as is done in the abstract. This mention should be followed by two or three sentences that contextualize the site and its relevance. The description provided in Supplementary Information 1 about the site generates more excitement to explore the results than the current introduction does.

2. Line 133 – The authors should provide more details about the design of their mtDNA probe set. It appears to be a new array not identical to the reference by Tejero et al., which is cited in the supplement for probe set design. Tejero et al.'s reference includes 54 mtDNA sequences, while this set contains 51. Although I found the list of species in Tejero et al.'s set, I could not locate the specific probe design strategy.

3. Lines 357-358 – The authors state: "All these Fournol-ancestry individuals have haplogroup U2'3'4'7'8'9, restricted to Southern Europe and may represent an early population of humans with Fournol genetic ancestry in Western Europe."

o The term "Fournol-ancestry" needs to be clearly defined, as its meaning is not immediately apparent to all readers.

o The sentence suggests that this haplogroup is restricted to Southern Europe, but Posth et al. (2016) shows haplogroup U2'3'4'7'8'9 throughout Europe, including as far east as Russia. Please clarify if there is a misunderstanding.

o The phrase "early population with Fournol genetic ancestry in Western Europe" is unclear. Are Southern and Western Europe considered mutually exclusive in this context?

4. Lines 363-364 – The authors state: "So far, the Haplogroup U2'3'4'7'8'9 has only been identified in four individuals (Malalmuerzo, La Riera, Rigney, and Paglicci108)."

o Posth et al. (2023) lists 12 individuals with this haplogroup, including a sample from Poland. This contradicts the claim that the haplogroup is restricted to Southern Europe. Please verify or clarify this statement.

5. Line 380 – The statement "The finding supports the archaeological evidence..." should be revised. The quantity of mtDNA sequences retrieved should not be used as a direct proxy for past species abundance at the site. Less human aDNA might reflect a lower spread of human DNA rather than a smaller population. It is misleading to make such a statement without knowing the exact source of aDNA for all animals.

Minor Points:

1. Line 74 – "Assess" is spelled incorrectly.

2. Line 119 – There is a comma after the parenthesis instead of a period.

3. Line 123 – I recommend replacing 'undocumented genetic affinities' with 'undocumented genetic phylogeny,' as 'affinities' seems too vague.

4. Line 237 – The mention of assembling mtDNA for "21 Cervus etc..." could be confusing, as one might think the number refer to a number of individual instead of a number of sample. It would be clearer to state that this refers to the number of samples, not individuals.

5. Lines 175-176 – The sentence "We did not recover endogenous human DNA from level 130, but traces of contamination (Supplementary Table 3, Supplementary Section 3)" should probably refer to Supplementary Table 9, where deamination rates are presented, as there are no human results in Supplementary Table 3.

6. Figure 1 – The sample ID numbers are not very visible unless zoomed in. Consider making the circles slightly larger and using bold numbers for better visibility.

7. Figure 2 – The color label for *Panthera pardus* is inconsistent between panel A and panel B.
8. Supplementary Material 2, Line 215 – The phrase "We applied contamination control measures to mitigate the effect of contamination" lacks specificity. Clarify the specific measures taken.
9. Supplementary Table 1 – The column "Sequenced reads" appears twice in the table.
10. Supplementary Table 7 – The first row corresponding to sample "El Miron_26" appears to have incorrectly reported numbers for some taxa, as the numbers are identical for *Capra ibex*, *Rupicapra rupicapra*, *Ovis aries*, *Bison bonasus*, and *Bos taurus*. Please verify the accuracy of these numbers.
11. Supplementary Figure 4 – Reporting R^2 values, as done in Supplementary Figure 3, would help readers assess the correlation between archaeological level age, relative abundance, and inhibition.

Reviewer #3

(Remarks to the Author)

Respect this manuscript, I have seen and evaluated the authors' comments and the modifications they have made, and from my perspective I agree with the modifications they have made.

Reviewer #4

(Remarks to the Author)

Version 2:

Reviewer comments:

Reviewer #2

(Remarks to the Author)

We have reviewed the initial submissions of the manuscript titled 'A sedimentary ancient DNA perspective on human and carnivore persistence through the Late Pleistocene in El Mirón Cave, Spain' by Gelabert et al., and we are satisfied with the current version.

The manuscript is clear and well-balanced, demonstrating the advantages of using sedimentary ancient DNA to complement traditional archaeological methods for studying past human and animal occupations at different points in time within a single location. It offers additional insights into the archaeology of El Mirón that would have been difficult to achieve otherwise. This work represents a significant contribution to future similar studies at other sites.

The authors have addressed all concerns, provided clarifications and added supplementary notes where requested by the reviewers. They approached the uncertainties carefully and made efforts to incorporate clarifications in both the main and supplementary texts as needed.

There are some biblio formatting discrepancies between the Word and PDF versions of the main text that have been submitted. Two references on line 121 of the Word file are not properly formatted, which created the discrepancies. I recommend the authors carefully review the references, as I also noticed, for example, that reference 40 does not appear to be relevant.

Overall, the authors' interpretations are logical, and their conclusions are robust. We do not detect any scientific flaws in the manuscript, and in my opinion, it is acceptable for publication.

Reviewer #4

(Remarks to the Author)

Response to reviewers

We thank the reviewers for their constructive comments. Our response is divided into two sections: the first details the main structural changes made, and the second addresses the reviewers' comments point by point.

1. Main changes made in the revised manuscript

- We have entirely rewritten the supplementary material, which is now more extensive and provides substantial information about context and experimentation. We now include a table and description with all the references to mitochondrial genomes used in the analyses.
- We have designed and carried out a new experiment that addresses the inhibition level present in our samples. Reviewers can find it in Section 2 of the Supplementary Information.
- We provide charts of the distribution of damage and coverage across the genome for all the recovered partial sedaDNA genomes. We also describe with precision how we have created these.
- We report a new partial genome and the phylogeny of *Equus sp.*
- We report a new phylogeny for *Cuon alpinus* that suggests that the diversity in El Miron sediments is more similar to one specimen from Bacho Kiro than to another *Cuon alpinus* sequence from Jáchymka cave, supporting the claims of at least two different mtDNA haplotypes of *Cuon* in Europe as suggested by Taron et al 2021 (*Taron et al. 2021*)
- We have removed the comments on the results before the species classification and reported statistical tests regarding species presence and sample performance. The results of the statistical tests all support all the current statements.

- We discuss the limitations of the present study in a new section of the manuscript discussion.

2. Point-to-point answers

Reviewer #1 (Remarks to the Author):

The article titled "A sedimentary ancient DNA perspective on human and carnivore persistence through the Last Glacial Maximum in El Mirón Cave, Spain" analyses various samples of DNA extracted from sediments of Miron Cave, showing interesting results regarding the presence of animal species never before identified in the cave through traditional zooarchaeological means. The manuscript also provides important data on the survival of some animal species, such as hyenas, until very advanced dates, which is one of the most intriguing findings of this study.

As a reviewer, I believe my role should focus, given the multidisciplinary nature of this study, on my specific area of expertise, which is zooarchaeological and taphonomic studies. My expertise does not extend to DNA studies, so I believe that aspect should be systematically reviewed by specialists in this field. Therefore, I will focus my comments to the authors on specific aspects of the archaeological work, and more particularly on the zooarchaeological issue.

Firstly, I'd like to say that I believe approaches like this are of great interest to archaeological work. In Zooarchaeology, we face the significant challenge that it's not always possible to taxonomically identify faunal remains. Often, we have to make inferences based on the size of the animal and the general characteristics of the assemblage. A clear case is the analysis of Bovidae in African sites. Thus, these types of approaches are very important and help us better understand the use of anthropogenic spaces and the paleoecology of archaeological sites. Therefore, I would like to start by congratulating the authors for their efforts and their interesting results.

However, I have some comments that I believe could help improve the content of the manuscript, not the formal study itself. I think where the manuscript falls short is in the presentation of the site, which is somewhat lacking. A publication in Nature Communications deserves attention to such formal matters, even if it requires repeating

information published in previous works, especially if it's information that can be included in the Supplementary Information (SI). Below, I will explain what I mean.

Firstly, I'm not entirely convinced that the title of the article aligns well with its content. I understand the focus on the Last Glacial Maximum, but the study analyses archaeological levels from the Middle Palaeolithic onwards. I think the title should more generally encompass the different chronologies addressed and not solely focus on the last period.

We have explored a new title and propose the new title: **"A sedimentary ancient DNA perspective on human and carnivore persistence through the Late Pleistocene in El Mirón Cave, Spain"** The reviewer's comment improves the former title and adequately addresses the topic of the paper.

The introduction seems generally correct to me. It is well-structured and presents the ideas that will be further discussed. However, I believe it's important to highlight, when discussing the site, that ALL the faunal information available comes solely from a 3 m² area of the site. If I understand correctly, both from the manuscript description and the Supplementary Information (SI), all DNA and zooarchaeological data from the site studies come from the excavation squares V, W, and X10. I think it's important to explicitly include this information in the manuscript. It's a small excavation area, and although it provides high resolution, it remains a very limited surface area.

In the revised version, we have provided a comprehensive description of all the sampled material in the newly expanded Supplementary Section 1. This section also includes extensive information about previous research conducted at the site.

We have also included the following sentence in **Line 129** of the main text: *"We screened 32 sediment samples (Supplementary Table 1) from the Rear Vestibule, of El Mirón cave (deep sondage meter-squares V, W, X10) (Figure 1, Supplementary Figure 1, Supplementary Section 1) spanning from the late Mousterian to the Initial Magdalenian (>46,000-21,000 cal BP)."*

We have included this information in the new section of the discussion that addresses the study's limitations. The reviewer can find in this section extensive debate about the possible results bias because of the studied location within the cave and the size of the studied pit. Line 447 of the main text "Firstly, our focus area within the vestibule of El Mirón is limited to 2-3 m², whereas the entire vestibule site spans over 300 m²."

In Figure 1A, I believe other sites in the area could have been included to provide a clearer context for the site. After all, it's just a grayscale map, and there are two Spanish cities that have little relevance to the study's theme. I encourage the authors to include a more elaborate physical map showing the location of Miron Cave and other caves in the area. Cities can be included as references, but I think the other option is more interesting. Furthermore, and I emphasize this point, I think efforts should be made to improve Figure 2B and combine it with Supplementary Figure 1 since a site plan is important for understanding the stratigraphy being presented. I reiterate, it's only 3 m², a small area, and it's important for the reader to be able to assess the results based on that information.

We present new Figures 1 and 2 that arise from the improvement of the former, we also present new panel 2B which shows the zooarcheological results from previous analyses. We now present the new Supplementary Figure 1, which is a combination of the former S1 and the new Panel S1B.

Regarding the plan, we have improved Figure S1 to help the reader locate the studied area within the cave.

Moving on to the results, but continuing with the same issue, in line 159, the authors say, "level 130 ONLY yielded 111 lithic artefacts." 159 artifacts may seem few compared to other sites, but the problem is that the size of the excavated area is not clear. Please remove the value judgment of "ONLY" and try to rephrase that sentence.

We deleted the word 'only' from the text. We provide context that we only referred to the studied trench corresponding to the vestibule rear, the north section of El Mirón cave (squares V, W, X10). We would like to remark that all the sampled levels come from the same area therefore we can compare the archaeological information across levels

We have added the following sentence in line **160**: “These Levels (130-121) are located at the rear of the cave (squares V-W-X10, about 2-3 m² of excavated area), (Figure 1B, Supplementary Figure 1)”

Lines 165-168. According to the authors, the Mousterian levels clearly have a lower density of archaeological remains than the Upper Palaeolithic levels. What explanation do the authors provide for this fact? Does level 130 have a greater sedimentary thickness than the rest, suggesting it was less utilized by human groups? Are there diagenetic issues that could explain a lower density of archaeological material in this level?

We are convinced that the lower amounts of archaeological material in level 130 are because of less intense human presence, in comparison to later periods, as described in the literature (Marín-Arroyo et al. 2020; Lawrence Guy Straus and González Morales 2012).

Levels 121-130 are described by Farrand et al. 2012 (Farrand 2012) as “colluvial terrace” material. Levels 130-121 are a series of colluvial, sandy loam deposits. Among these are finer grain levels, others more gravelly, with varying amounts of alluvial pebbles and cobbles and limestone rocks spalled rocks from the cave walls and ceiling, generally yellowish-brown or brownish-yellow, but a few browner or olive-brown. The lay-sized sediment fraction ranges from 30 to 50%, the rest is coarse fraction. There are no visible changes in this ratio through the sondage. There is no evidence of gullying or significant water disturbance, but the materials are quite scarce and dispersed through quite thick levels (especially 130)(Geiling 2020) reveal that the bones (including ones of very small animals—birds, lagomorphs) are in fine condition and showing not signs of transport (i.e., surfaces not altered by running water or other diagenetic processes, etc.) in these levels. So, neither the sedimentology nor the taphonomy would lead us to conclude significant disturbance. We do not see evidence of the deposits being in a secondary position.

We have included such information in the new Supplementary Section 1 of the revised version, in which we have also extended the information about sedimentation and archaeozoology.

And with this, I come to the main concern I have with this study. The work is conducted through the analysis of sediment samples from different levels. However, the geological context of the analysed sequence is not provided at any point. Only a figure of a field drawing with different colours depicting a three-meter-long section is included, but what is the grain size of the levels? What does the sedimentological analysis reveal? How were these levels formed? Why is there such a pronounced slope? Is this slope diagenetic, or is it attributed to the original topography? As you can see, I have quite a few questions about the geology just by looking at the section shown, which doesn't provide any information beyond the layering of levels. The authors may refer to previous publications to defend this, but I think it's mandatory to include a section in the Supplementary Information (SI) that thoroughly develops the stratigraphy of the analysed sequence. How can we trust the DNA results without being sure that the stratigraphic sequence hasn't been altered by hydrological, gravitational, or other processes? To exemplify this, we could talk, for example, about the case of El Pendo cave. As the authors may know, it's an important cave with a very long sequence, but recent geoarchaeological studies have shown that the sequence is mostly in a secondary position. If this same study were done in El Pendo cave without considering this data, could we trust the results? Consider this question as the main reason behind my inquiry.

As previously mentioned, we have included a new sub-section in the revised version (Supplementary Section 1) titled "Sedimentology and Sample Stratigraphic and Taphonomic Context." This section provides a comprehensive description of the state-of-the-art in cave formation and sedimentology. We believe this addition meets the reviewers' requests for more detailed context.

El Mirón Cave, features an extensive sequence of dated materials, offering high-resolution support for the depositional history of the site. According to the previous research, described in the new section, no indications or observations are suggesting that sediments have been displaced across different levels. The available analyses support

that the current stratigraphic profile represents the depositional history of the sequence in the sampled area.

Regarding the reviewer's concern about the slope near the sampling location, there is no evidence of bone or material movement in the strata located close to the slope. The slope is part of the original topography. The bones from the studied stratum do not show signs of being washed, indicating minimal impact from water flow. This suggests that the stratigraphy of El Mirón remains intact without visible disturbances.

Lastly, as a final argument for the lack of DNA movement across the sequence, we would highlight that the DNA results from all examined taxa align with the known genetic lineages of the Pleistocene, further corroborating the temporal integrity of the sediments.

Following the reviewer's suggestion, we have also included the following statement to discuss other cave stratigraphic problems in Supplementary Section 1 " Similar studies at other Cantabrian archaeological sites have revealed secondary depositions and disturbances. For instance, in La Cueva del Pendo (Sanguino and Montes 2001; Montes et al. 2005), a taphonomic reappraisal of the sequences suggests that the interstratifications do not correspond to a primary archaeological sequence but may result from post-depositional processes. A similar situation has been reported at La Güelga cave, where micromorphological observations and new radiocarbon dating strongly suggest that the few presumably Châtelperronian finds were transported (Kehl et al. 2018)."

Similarly, I believe it would be interesting to include not only a drawing of the analysed section but also some photographs. Additionally, I'm sure the authors have photographs of the sediment sampling process, so it would be good to include some in the SI.

We have included photographs in the revised version. The reviewer can observe these in Supplementary Figures 1 and 2, which depict the sampling process and the sampled section. This section is now better described in the previously mentioned Supplementary Section 1. We have also improved Figure 1.

The authors have already published some works on this site, including Geiling's embargoed thesis. I think it would be good, in addition to the geology, to include a section in the SI that develops the archaeological part of the site in a slightly more detailed way.

It doesn't need to be overly developed, but I think it would be good to have a paragraph for each level detailing the types of lithic industry and fauna present in each level, as well as the type of human and carnivore activity identified in the levels (are they campsites? Carnivore dens?). It would also be helpful to include tables with more explicit numbers of remains. For example, the Minimum Number of Individuals (MNI) of the fauna, I believe, is very important to include in this study. Only the NISP value has been shown in the supplementary tables, and I think it's a rather poor piece of data that doesn't do justice to the extensive work done with the DNA data. Please improve the archaeological part in the SI; it will be a way to highlight the significance of the site and help readers understand the importance of the data, avoiding the misconception that because some of this data is already published, it's not worth including here. Nature Communications is a multidisciplinary journal, and therefore it's important that the archaeological part is also introduced, as it will help understand how the site functions.

In conclusion, I believe this is a good study, but I think the manuscript and SI fail to address a basic aspect: the context of the site. We could debate whether archaeological data are necessary or not, but in the case of geological data, I believe they are **MANDATORY** in this study because without them, we cannot validate the genetic data.

We have extended Supplementary Sections 1 and 2 to include a comprehensive description of the previous archaeological and zooarchaeological work. Additionally, we have incorporated the MNI information in Supplementary Table 5. We hope the reviewer finds this version improved. We have included a new Table 5, which compares the MNI to the NMISP, and provided detailed information regarding the NSIP in Figure 2. As mentioned before the geological data has also been extensively incorporated in Supplementary Section 1. In addition, we also present analyses that compare the NISP and sedaDNA results (Supplementary Tables 15-17)

Reviewer #2 (Remarks to the Author):

Review for “A sedimentary ancient DNA perspective on human and carnivore persistence through the Last Glacial Maximum in El Mirón Cave, Spain» from Pere Gelabert et al.,

Gelaber et al. used animal mitochondria targeted sequencing to retrieve aDNA from past animals from sediment samples collected from the lower archaeological stratigraphic layers of El Miron (Spain). The novel contribution of this manuscript is the reconstruction of animal occupation history at El Mirón Cave during the Pleistocene using ancient DNA retrieved from sediments. The authors recovered ancient DNA from 29 animal species, about half of which were not known from the skeletal record. The authors provide evidence that some species were present for a longer time than what is known from the skeletal record, including some that were believed to have gone extinct earlier. Mitochondrial genomes were reconstructed for several species, allowing the authors to place them on a phylogeny. The genetic relationship to species outside El Mirón Cave permit inferences about relatedness across Eurasia and broaden the scope of the manuscript. However, while the data generated is interesting, the study's framing makes it difficult for the reader to understand the main point of the manuscript, what the author intends to convey, and what we can truly learn from this new data. The paper appears to be a methodological study but also shows promise in investigating interactions between different species during the Upper Paleolithic in the region, animal occupation at El Mirón Cave, and human mtDNA continuity in the area. While many of these points are suggested by the data, none of them have been thoroughly investigated, and at times, evidence supporting claims is not clearly reported, leading to some interpretations of the results sounding naive and the overall study appearing somewhat superficial. Overall the work and the data generated are interesting but the paper still requires major revisions in order to show how relevant these data are in a methodological and biological context. I suggest to the author to reframe the entire paper as a methodological study on using sediment ancient DNA to investigate the faunal occupation of an archaeological site, using El Mirón Cave as a case study, and include the other biological insights as peripheral outcomes. Of course, this suggestion is optional, provided that the authors finds a better angle and produce a more robust paper.

We have revised sections of the paper to make them more robust. However, we would like to maintain our original narrative and avoid turning it into a methodological paper. We have, however, significantly written the results and discussion sections. We are convinced that the current version strengthens the main results of our work.

General major comments

1. The lack of numbered lines makes it difficult for the reviewer to pinpoint specific sections.

We add numbered lines in the revised version.

2. Each paragraph of the introduction seems to address a completely different study, lacking clear continuity and making it less engaging.

We have rewritten the introduction to make it more compelling by restructuring some paragraphs and removing redundant information. We are confident that the current introduction reads more effectively.

3. There is too much discussion in the results section. I would suggest to the authors to either combine the Results / Discussion in one section or clearly report the results with some context in the results section, saving the detailed interpretation for the discussion section. Currently the discussion section is not worth reading as it merely summarizes what has already been discussed previously, for instance the first paragraph of the discussion is very similar to the first paragraph of the introduction.

We have rewritten the discussion and results section to relocate concepts that are relevant to the discussion. We are convinced that the current manuscript is improved.

4. One of the major issues with the paper is that the authors don't discuss any potential limitations of their approach. Inferences using ancient DNA from sediment should address potential of DNA translocation between layers. The authors should contextualize

the results concerning the integrity of the stratigraphic layers and discuss how consistent the results are in comparison.

We have addressed this issue in a new section of the discussion, where we elaborate on the limitations of our work related to the archaeological conditions and methodology. As mentioned in a comment from Reviewer 1 we have added substantial information in SI section 1 regarding the geology and cave description.

We have added the following text in the discussion (line 446) “We have identified several limitations in our study that could impact our findings. Firstly, our focus area within the vestibule of El Mirón is limited to 2-3 m², whereas the entire vestibule site spans over 300 m². This specific focus, aligned with previous studies on the lower sequence of El Mirón may bias our results towards this particular area and its materials. This limitation suggests that different regions within the vestibule could yield different outcomes. Moreover, the location of our study area at the rear of the vestibule might influence the higher presence of carnivore DNA. A larger cave sampling, especially in the front vestibule 49,81, may produce more robust statistical data. In addition, future research microcontextual studies should be carried out to integrate the current results with the microstratigraphic context of the sediments at El Mirón⁸², delving into the specific origin of the DNA and resolving possible secondary alterations, that have not been identified so far with the current sedimentological, taphonomical, geological and chronological research at El Mirón^{48,49,83–86}. Secondly, linking the amount of DNA discovered to specific activities or animal presence is challenging due to uncertainties regarding the tissue or organic material origins (i.e., faeces, saliva, urine, or other fluids), our data proves the presence of the notorious amount of DNA from species without remains, further studies on DNA tissue origin can link animal activities and DNA presence. Third, we stretched the capacities to study ancient genomes due to the small amount of available mtDNA Pleistocene genomes of relevant species such as dholes or leopards, this limits our analyses and the inferences we can make. Finally, a potential limitation lies in the risk of the presence of DNA not contemporaneously deposited with the sediment. However, our rigorous validation methods and comprehensive analyses are designed to address and minimize these concerns effectively, we have evidenced the capacity to exclude the possible contaminant taxa and verify individually the identifications. The genetic

similarity of our data to other Pleistocene sequences and the temporal dynamics strongly support that the DNA is contemporaneous with the stratigraphic levels and available dates. “

5. Another important issue, is that the authors seem to interpret the amount of sequencing reads as an indicator of specie abundance at the site. This would assume that all the animals that were present at the site left equivalent traces of DNA relative to the time they spent, which is not true. Megafauna body decomposition at the site is likely to leave more DNA than a few human individuals just visiting the sites. Also, layers can have different characteristics leading to different property of aDNA preservation. Even in the same layer, the preservation can be different. A layer more conducive for aDNA preservation will generate more aDNA than a layer that is less. One cannot use amount of aDNA preservation as indicator of species abundance.

We have explored the topic of abundance in the new Supplementary Section 3, where we test the relationship between NSIP and sedaDNA. Our findings indicate that while sedaDNA can reflect the presence of prey animals, it does not consistently do so for other species. This point has been addressed in the main text (Figure 2 and Supplementary Section 3). We mention the following statement in the main text (line 172) “Overall, the distribution of prey taxa is in agreement with the archaeofaunal descriptions. However, this is not the case for carnivores and uncommon taxa (Figure 2, Supplementary Section 3)”.

Additionally, we have removed the comparison of animal quantities across levels. Quantity is only used to test the similarities between sedaDNA and the archeofaunal register, laying the groundwork for future tests.

As a result, Figures S2 and S3 from the previous version have been removed, and Figure 2 has been reframed accordingly.

Specific major comments

1. The results presented in Figure 2 should be adjusted for sequencing depth before making claims about differences in species abundance between layers, as this could contribute to differences in the number of reads obtained per species.

We have now improved Figure 2 and corrected it by sequencing the depth. We also incorporate the NISP in Figure 2. In this direction, we have also included major modifications regarding the correspondence of sedaDNA and NISP in Supplementary Section 3.

2. The authors claim the results presented in Figure 2 are congruent with the archaeological findings as the Mousterian horizon had the lowest average read number for all species compared to the upper layers (lines 165-167). However, the results are also congruent with the preservation of ancient DNA over time, as less DNA is expected to be preserved in deeper layers than upper layers. No estimates of inhibition are provided, which could also contribute to differences in ancient DNA preservation across layers. These two alternative explanations are congruent with the results and should be mentioned in the manuscript. The claim also appears to be contradicted by the results presented in Supplementary Figure 2, where the average read number is highest in Mousterian layers for humans.

We have investigated the possibility of inhibition and designed a new experiment, which we include in Supplementary Section 2. We observe that inhibition is present in major intensity where DNA is present in major intensity pointing out a relationship to the overall organic material preservation. Therefore we evidence that the lower DNA yields in level 130 are due to poorer organic preservation. We have added this text in the main text “Based on a PCR assay, we tested the inhibitory effect of the purified extracts, and we found that the lower DNA yields of level 130 are not related to inhibition and are likely due to poor organic preservation (Supplementary Section 2, Supplementary Figures 3-4).” (line 140).

We want to emphasise that the human DNA found in Mousterian layers can only be explained by modern contamination (marginally, probably with environmental or old excavation campaigns). This data is presented in Supplementary Table 3 and described in Supplementary Section 4 (subsection 12). “Specifically, we detected the presence of human reads in all samples, but only 10 had deamination values greater than 30%, which we consider to represent genuine Pleistocene DNA. None of these samples correspond to level 130, indicating that there is no evidence of human DNA linked to the Mousterian culture in El Mirón”.

The exact sentence in the main text referring to this is “We did not recover endogenous human DNA from this level, but traces of contamination (Supplementary Table 3, Supplementary Section 3)” in line 175.

3. The authors should discuss if the results obtained from euka reported in Supplementary Figures 2 and 3 can be used to reliably support some claims (lines 172, 174). The results obtained with euka for humans do not appear to hold up after further processing the assigned reads and then mapping to the reference genome. For example, 6932 reads are assigned by euka to humans in Mousterian layers (Supplementary Table 2), some of which are presumed to be ancient given the deamination values set as priors (Supplementary Section 1), but 0% appear to display signs of deamination after mapping to the reference genome (Supplementary Table 8).

We agree with the reviewer that the above-mentioned problem can show consistency problems. In the revised version, we have used only the reads classified at the species level to make such claims. We use euka to classify reads at the family level and these are later individually checked with stringent thresholds (deamination and length). We have now rewritten the results, and we only talk about the validated reads with the mapping approach in all the text. We have, therefore removed Figures S2 and S3 of the former submission. We only mention species or genera that show clear signs of being ancient.

This specific process is mentioned in Supplementary Section 3 (line 340 of the SI) with the following text:

“This classification process enabled us to distinguish the animal families present at El Mirón and prepare the data for the single-taxa analyses (Supplementary Table 2). We do not use this data to assess the relative presence of taxa in the samples due to classification limitations and the inability to specifically validate these reads. “

4. The claim of an increased presence of carnivores such as *Panthera pardus* in layer 126, which could indicate less human and other animal presence linked to the coldest period of the LGM (lines 173-175) do not appear to be supported by the data. Layer 126 appears to be the peak of human presence across all layers and there is no increased presence of *Panthera* (Figure 2).

We have rewritten the section as mentioned in the previous comment, ensuring that all statements regarding animal presence align with the reported data. The specific comment about the differential presence of carnivores has been removed. This is described in the text with the following statement in line 202: “We have identified an average of 3.4 species of carnivores per sample, with 8 samples from levels 129 to 119.2 showing up to 5 different species in a single sample (Supplementary Tables 3 and 5). Previous archaeozoological research on El Mirón levels 130-119.2 identified *Canis lupus* fragments only in levels 128 and 130 and no *Cuon alpinus* fragments are reported (Marín-Arroyo et al. 2020). In contrast, here we identify the presence of both canids (dhole and wolf) in all the studied levels of El Mirón (Supplementary Tables 3-4, Figure 2). Finally, based on the study of sedaDNA, we determined that while no cave bear (*Ursus spelaeus*) is present in El Mirón, genetic traces of brown bears (*Ursus arctos*) are present throughout the lower profile (Mousterian-Initial Magdalenian) of the site sequence.”

As we mentioned before we no longer compare the proportion of a species across levels as we discussed that there is not enough sample to do this without bias.

5. Results supporting the claim that "the presence of elevated carnivore sedaDNA can be used as indicative of low-intense human occupation" (lines 387-389) should be provided. It's difficult to assess exactly which results, and the number of observations, support this claim. This claim warrants explicit evidence, given that human and large carnivore competition is a main section of the introduction.

We have rewritten this comment as a proper examination did not support this claim (line 400)

“ We report the presence of carnivores (*C. crocuta*, *L. pardinus*, *P. pardus*, *C. lupus*, *C. alpinus* and *V. vulpes*) across all stratigraphic levels, with no evident differences in the number of carnivore species per sample except for notably lower preservation in level 130. Notably, levels 128-130 are situated near the cave walls in the rear part of the vestibule, an area closer to the dark inner cave. This location likely attracted carnivores more frequently than humans, as indicated by the limited archaeological evidence of human occupation in this part of the cave as observed in Amalda cave.(Sánchez-Romero et al. 2020) In the future, denser sampling in space and time could provide substantial data to identify quantitative differences, resolve occupation hiatuses, and date alteration patterns. We also observed that older samples exhibit reduced DNA preservation, which is unrelated to inhibition. Additionally, our data suggest that the archaeofaunal representation aligns more closely with sedaDNA findings for prey animals rather than carnivores. The persistent presence of carnivore DNA, coupled with their absence in the physical remains, indicates that carnivores frequently utilized the cave during the Palaeolithic. They likely scavenged leftovers from humans and intermittently occupied the cave in the absence of humans (Sanchis et al. 2019). While carnivores may leave biological traces such as faeces and urine, their physical remains, such as teeth or bones, are less commonly found unless they perish in the cave. Future research linking sedaDNA to tissue origin might shed critical light on cave occupation patterns. ”

As mentioned at the beginning of this document, all the current statements are supported by the reported statistics results.

6. While the phylogenies of ancient mitochondrial genomes permit the authors to place their results from El Mirón in a broader geographic context, the discussion of the results is lacking with little inferences drawn or interpretations offered. For example, the phylogenetic results of *Panthera pardus* depict the genomes from El Mirón being more closely related to the genome from Mezmaiskaya Cave in Russia than the genome from nearby Baumannshöhle in Germany (Figure 4). The authors say this finding is surprising (line 422), but don't explain why or provide an interpretation of this result.

We have added the following sentences (line 285)“ *P. pardus* is the only species from the genus *Panthera* identified at El Mirón through archaeological faunal analyses. Our sedaDNA analysis confirmed this result. We recovered two partial genomes of *P. pardus* from levels 129 (46890–33160 cal BP) and 125 (22980–22240 cal BP), showing nucleotide diversity that suggests multiple individual origins. We generated a Maximum Likelihood tree that shows that both El Mirón sequences form a clade closely related to the BAR001 genome from Mezmaiskaya Cave (Russia, Northern Caucasus)(Paijmans et al. 2018) (Figure 4, Supplementary Figure 19). Hence, the El Mirón mtDNA consensus sequences are more similar to 35.000 YBP sequences from the Caucasus than the Baumannshöhle (Germany, ~ 40 kya) genomes, meaning that the *P. pardus* from Europe would not be a monophyletic clade providing further complexity to the scenario suggested in Paijmans et al. (Paijmans et al. 2018) “

7. There is also little interpretation provided regarding the phylogenetic results of *Canis lupus*, which depict different mitochondrial lineages that alternate in time at El Mirón, some of which are more closely related to those from Goyet Cave in Belgium (Figure 3). Are these faunal turnover events mentioned in the Discussion sections?

We do not consider this to be a turnover event; rather, our evidence indicates the presence of multiple lineages coexisting simultaneously. Additionally, we would like to highlight that this study represents the densest sampling of Pleistocene *Canis lupus* mtDNA from a single site to date, underscoring the importance of deep sampling. We have elaborated on this point in the results section (line 252)

“Only seven bone fragments of *Canis lupus* have been identified, where sedaDNA levels have been studied (Supplementary Tables 4-5). In contrast, all the levels except 119.2 and 130 yielded enough mtDNA to reconstruct partial *C. lupus* mitogenomes with average coverages ranging from 5.3X to 35X, These partial mitogenomes represent the densest up-to-date sequencing of mtDNA Pleistocene *C. lupus* of Iberia. The mitogenomes were aligned with Palaeolithic and modern *C. lupus* mitogenomes(Bergström et al. 2022, 2020; Thalmann et al. 2013; Skoglund et al. 2015). First, we observe that wolf lineages are diverse in time and space, all showing phylogenetic relationships with Pleistocene wolves

of Europe (Loog et al. 2020; Thalmann et al. 2013; Bergström et al. 2022). Some El Mirón sequences cluster close to Palaeolithic wolves from Goyet Cave, Belgium (Skoglund et al. 2015) (Figure 3, Supplementary Figure 11). The oldest available mtDNA sequences of Pleistocene European dogs fall close to or within dog clades A, C and D (Thalmann et al. 2013; Bergström et al. 2020; Hervella et al. 2022; Boschini et al. 2020). All the newly reported sequences from El Mirón fall out of this diversity and are close to the oldest wolves of the dataset (Figure 3). Therefore, we can only confirm the presence of wolves in the Solutrean sequence and not the presence of domestic *C. lupus* lineages (Thalmann et al. 2013). ”

8. The discussion of the human mitochondrial phylogeny results (Figure 5) requires more context to understand the significance of the authors' findings. The term "Fournol" is used (line 351, 353) without defining it or providing any context. I find it difficult to follow the meaning of the sentence "All these Fournol-ancestry individuals have haplogroup U2'3'4'7'8'9, restricted to Southern Europe and may represent an early population of humans with Fournol genetic ancestry in Western Europe" (lines 351-354). More explanation is needed to justify the claim that this haplogroup is restricted to individuals from Southern Europe and how it represents an early population of humans with Fournol genetic ancestry in Western Europe, perhaps in the Discussion section. The La Riera individual mentioned in the results section is not labeled in Figure 5.

We have included all available individuals with sufficient coverage in the phylogeny. The individual from La Riera lacks adequate coverage to be included in the phylogenetic analysis, and only the mitochondrial haplogroup has been reported (Posth et al. 2023) (Posth et al. 2023).

We have included the following statement (line 368) "So far the Haplogroup U234789 has only been identified in four individuals (Malalmuerzo, La Riera, Rigney and Paglicci108) (Villalba-Mouco et al. 2023; Posth et al. 2023; Fu et al. 2016). All these individuals are dated around the LGM or post-LGM and are from Southern or Western Europe. ”

Specific minor comments

1. Figure 1: The number of samples collected from the layers as depicted by dots in Figure 1 does not always match the number of samples reported per layer in Supplementary Table 1 (e.g. layers 126, 128, 129)

The number of dots has been corrected. Layer 126 was sampled two times, but an additional extract of one of the samples was analyzed. We have clarified this situation in the text (line 222 of the SI): 'Sample El Miron_18 was extracted twice from two different 50 mg aliquots to increase the chances of retrieving human DNA. ' This sentence is in point 4 of Section 2 of the supplementary material.

2. Figure 2: Figure legends says species are depicted, but three labels are at the genus level (*Rupicapra*, *Bos*, *Panthera*). Species names should be in italics.

We have redone Figure 2, including these comments. The names displayed the species names in italics.

3. Figure 3: Figure presents results from *C. lupus*, but the figure caption incorrectly references Supplementary Figure 10 which is a Caprinae phylogeny

We have corrected the citation

4. Supplementary Figure 2: Figure legend overlaps some text of the figure

We have removed this figure from the resubmission following reviewer 2's suggestion to limit the abundance usage for comparing presence.

5. Supplementary Figures 2 and 3: It would be helpful if the colors of the family names matched in these figures so that the reader can more easily compare the results.

We have removed this figure from the resubmission following reviewer 2's suggestion to limit the abundance usage for comparing presence.

6. Supplementary Section 2: It's reported that mammoth (*Mammuthus primigenius*) and woolly rhino (*Coelodonta antiquitatis*) DNA was retrieved from layer 130, but no data is reported in Supplementary Table 2 supporting this claim.

We have reanalyzed these two identifications and now we provide the credible identifications in Table S3 and the discussion.

7. Supplementary Tables: Sample "El Mirón_18" is listed twice with different results, but no explanation why.

These refer to two samples from the same sediment but from two extracts. This process was carried out in order to maximise the obtained human DNA. We explain this particularity in Supplementary Section 2.

8. Methods: Incomplete first sentence (line 457) after the "Sampling" section

We have corrected the sentence: "Sampling was performed in the profile of the archaeological excavation trench at the rear of the El Mirón vestibule in excavation units V-W-X/10 and the W-X10 deep sondage (Figure 1)(Lawrence G. Straus and González Morales 2018)"

9. Methods: No information was provided about the reference genomes used for mapping with bwa

We have included this information in the Supplementary section 4.

10. Methods: It's stated that "We determined the presence of partial genomes when the average coverage of a taxon was >5X" (lines 476-477). However, the human mitochondria genome sequence retrieved from El Mirón_1 only has a coverage of 2.5X and is included without explanation.

We have considered this situation an exception. The coverage of 2.5X has enabled us to place Mirón_1 within the diversity of the other two samples.

11. There are differences in terminology used throughout the manuscript (e.g. "levels" vs "layers"; "euka" vs. "EUKA"; "genomes" vs. "sequences"). Perhaps the authors could chose one term if they have the same meaning.

The terminology has been corrected and now appears as consistent.

Reviewer #3 (Remarks to the Author):

This manuscript entitled “A sedimentary ancient DNA perspective on human and carnivore persistence through the Last Glacial Maximum in El Mirón Cave, Spain”. It is a very interesting and relevant study that shows how useful it is to apply new analysis techniques to the study of archaeological sites, since through sedimentary ancient DNA the presence of species that had not been documented from a fossil perspective can be documented. . Just because of this type of evidence, I believe that this article represents a new advance in the discipline, and although we know this type of analysis exists in archaeology, it has rarely been able to be applied with the success that has been applied here. This study highlights the excellent results that the Miron team has been achieving in recent years. However, although the research results are important, there are still aspects that the authors could develop in this article.

We thank the reviewer for the comment.

On the one hand, the authors analyze the presence of carnivores and humans based on DNA data. The study reveals the new presence of various carnivores, but new data on herbivores has also been observed. Given that the authors discuss subsistence strategies in the Middle and Upper Paleolithic, it would have been interesting for the authors to develop in one or two paragraphs what implications the recovery of DNA from herbivores that had not previously been discovered at the fossil level has. Would it be possible for humans to transport meat or skin from animals that have not left bone remains? This would be very relevant data, since it would imply transport strategies for possible prey, in the form of fleshed meat or skin

We have included the following sentence (line 411) in the manuscript addressing the comment of the reviewer: “Additionally, our data suggest that the archaeofaunal representation aligns more closely with sedaDNA findings for prey animals rather than carnivores. The persistent presence of carnivore DNA, coupled with their absence in the physical remains, indicates that carnivores frequently utilized the cave during the Palaeolithic. They likely scavenged leftovers from humans and intermittently occupied the cave in the absence of humans (Sanchis et al. 2019). While carnivores may leave

biological traces such as faeces and urine, their physical remains, such as teeth or bones, are less commonly found unless they perish in the cave. Future research linking sedaDNA to tissue origin might shed critical light on cave occupation patterns.”

In our manuscript, we provide a more detailed analysis of the species present in El Mirón through sedaDNA data, demonstrating that it is possible to classify DNA from archaeological remains at the species level (Supplementary Section 3). Our findings include identifying carnivore species previously undetected in El Mirón, such as the Dhole. However, we do not report any new herbivore genera that were not previously identified through archeofaunal analyses, except for the wholly Rhino. We report that the archeofaunal remains and sedaDNA show more similarity for the herbivore species than for the uncommon carnivore taxa (Supplementary Table 16).

Minor commentes

Fig 2 is not well understood, I believe that this figure could be complemented with a table that shows the NISP and MNI of the taxa found in the Miron sequence by levels. In this way the information would be complete and easily visible to the authors. .

The figure has been redone.

Line 64. Authors can cite more appropriate references e.g. This quote from Lindly (1988), the work of Jones & Carvalho (2023)

Lindly, J., 1988. Hominid and Carnivore Activity at Middle and Upper Paleolithic Cave Sites in Eastern Spain. *Munibe Antropologia - Arkeologia* 40, pp. 45–70

Jones, E.L., Carvalho, M., 2023. Ecospaces of the middle to upper paleolithic transition: the archaeofaunal record of the iberian peninsula. *J. Hum. Evol.* 177, 103331 <https://doi.org/10.1016/j.jhevol.2023.103331>

Line 68 More appropriate than ref 2 that is cited, which is very generic, would be the ref by Pérez Ripoll et al (2009), which is more specific and specific to the Iberian Peninsula.

Pérez Ripoll, M., Morales Pérez, M., Sanchis Serra, A., Aura Tortosa, J.E. & Sarrión Montañana, I. (2010). Presence of the genus *Cuon* in upper Pleistocene and initial Holocene sites of the Iberian Peninsula: new remains identified in archaeological contexts of the Mediterranean region. *Journal of Archaeological Science*, 37: 437-450

We have changed it, the reviewer can identify it in the first paragraph of the introduction (line 68).

Line 65. Ref 5 is not appropriate for this context as it refers to an Olduvai Gorge context that is unrelated to the topic discussed in this introduction. More appropriate may be any reference that shows alternation between humans and carnivores in the occupation of the sites, for example the recent publication by Pinto et al (2024) or Amalda (Yravedra et al 2011), the quotes from Llonin (Sanchis et al 2019). and others

Pinto et al 2024. Alternation between humans and carnivores in the occupations of the Mousterian site of Sopena ~ rock-shelter (Asturias, Spain). *Quaternary Science Reviews* Volume 328, 15 March 2024, 108468

Yravedra, J., 2011. A taphonomic perspective on the origins of the faunal remains from Amalda Cave (Spain). *J. Taphonomy* 8 (4), 301–334.

Sanchis, A., Real, C., Sauqu´e, V., Nunez-Lahuerta, C., Egeuz, N., Tormo, C., Perez Ripoll, M., Carrion Marco, Y., Duarte, E., Rasilla, M. de la, 2019. Neandertal and carnivore activities at Llonin cave, Asturias, northern iberian peninsula: faunal study

of the mousterian levels (MIS 3). *Comptes Rendus Palevol* 18 (1), 113–141. <https://doi.org/10.1016/j.crpv.2018.06.001>.

The references have been incorporated. The reviewer can find it (Pinto-Llona, Yravedra) in line 65 of the Manuscript. Sanchis et al 2019 is cited in line 104 of the manuscript

Ref 31 Change Altuna 1971 to Altuna 1972

We have changed the citation to Altuna 1972

“37. Altuna, J. Fauna de mamíferos de los yacimientos prehistóricos de Guipuzcoa. Con catálogo de los mamíferos cuaternarios del Cantábrico y del Pirineo Occidental. San Sebastián, Sociedad de ciencias naturales Aranzadi 24, 1–465 (1972).”

Line 109-112: The authors do not include citations referring to the presence of some large carnivores existing in the Cantabrian region. For the Magdalenian there are records of *Panthera pardus* in the sites of El Castillo, El Juyo (González Echegaray & Freeman, 2015). For *Panthera leo* there is a citation in Abauntz (Jerjotoma-Ortín et al 2024), el Juyo (see González Echegaray & Freeman, 2015) or La Paloma (Castaños, 1980), Cuon in Rascaño 4 (Altuna 1981). Which complement some of the quotes referring to sites such as Urtiaga. Mentioned in the text.

Jerjotoma-Ortín et al 2024 The Mark of the Beast: a bone assemblage assessment from the North of the Iberian Peninsula (MIS 3). *Journal of Archaeological Science: Reports* 54 (2024) 104409

Altuna, J. (1981). Restos óseos del yacimiento prehistórico de Rascaño (Santander). In (González Echegaray, F. & Barandiarán, I. eds.) *El Paleolítico Superior en la Cueva de Rascaño*. Santander, pp. 221-269

Gonzalez Echegaray & Freeman 2015. Excavando la cueva del Juyo un santuario de hace 14000 años . *Monografías del Museo Nacional y Centro de Investigación de Altamira*, n.º 25

Castaños, P. 1980 LA MACROFAUNA DE LA CUEVA DE LA PALOMA (PLEISTOCENO TERMINAL DE ASTURIAS). En Hoyos M. et al 1980. *La Cueva de la Paloma*. Ministerio de Cultura.

We have included the suggested citations. The reviewer can identify these citations in the third paragraph of the introduction (lines 94-110)

Line 143-145. The authors indicate that they have been able to identify 51 species and refer to the supplementary files. However, these 51 species are not clearly seen in these files. As a suggestion, it would be important for the authors to put in a table which species are recognized at the fossil level and which are identified at the DNA level. It can be indicated as a presence/absence table.

The number 51 refers to the species included in the capture design, which is the tool we use to recover DNA from metagenomic samples. We do not recover DNA from 51 species. We state in the text that we detected the presence of 31 species (line 134 of the Manuscript) “. After a strict classification process (Supplementary Section 2), we detected the presence of 31 species of animals (including humans) with congruent signals of aDNA based on length and deamination (Supplementary Tables 2-3, Supplementary Figure 4, Supplementary Section 2).” To clarify this, we have extended the section referring to the capture kit in the Supplementary Section1

Line 180-182: The authors cite refs 26, 28, 45-47, however these references are not the most appropriate. Ref 26-28, 45 refer to the extinction of some carnivores. The ref. 46 refers to a paleontological site without human presence. There are multiple Cantabrian sites with alternating humans, carnivores and bears in the occupation of the sites. Labeko koba Amalda, Abauntz, Llonin, Esquilleu, Sopeña, Ekain, la Viña, Castillo, etc.

Sanchis, A., Real, C., Sauqu´e, V., Nunez-Lahuerta, C., Eguez, N., Tormo, C., Perez Ripoll, M., Carrion Marco, Y., Duarte, E., Rasilla, M. de la, 2019. Neandertal and carnivore activities at Llonin cave, Asturias, northern iberian peninsula: faunal study of the mousterian levels (MIS 3). *Comptes Rendus Palevol* 18 (1), 113–141. <https://>

Torres-Iglesias et al 2024

Pinto-Llona et al 2024

We have included new citations in this section. The introduction has been rewritten; The reviewer will find that all the suggested literature is now cited in the introduction.

Torres-Iglesias 2024 is reference 18

Pinto-Llona 2024 is reference 5

Sanchis 2019 is reference 14

Line 377: The authors refer to two zooarchaeologic and taphonomic studies of the transition between Neanderthals and anatomically modern humans in the Cantabrian Sea in relation to the activity of humans and carnivores, but they do not cite other works developed by other authors for the same region that also are relevant in this relationship between humans and carnivores eg Yravedra (2011, 2013), Pinto et al (2024), Sanchis et al (2019)

Sanchis, A., Real, C., Sauqu´e, V., Nunez-Lahuerta, C., Eguez, N., Tormo, C., Perez Ripoll, M., Carrion Marco, Y., Duarte, E., Rasilla, M. de la, 2019. Neandertal and carnivore activities at Llonin cave, Asturias, northern iberian peninsula: faunal study of the mousterian levels (MIS 3). *Comptes Rendus Palevol* 18 (1), 113–141. <https://doi.org/10.1016/j.crpv.2018.06.001>

Yravedra, J., 2011. A taphonomic perspective on the origins of the faunal remains from Amalda Cave (Spain). *J. Taphonomy* 8 (4), 301–334.

Yravedra, J., 2013. New contributions on subsistence practices during the Middle–Upper Palaeolithic in northern Spain. In: Clark, J., Speth, J.D. (Eds.), *Zooarchaeology and Modern Human Origins: Human Hunting Behavior during the Later Pleistocene*. Springer, New York, pp. 77–95

Pinto et al 2024. Alternation between humans and carnivores in the occupations of the Mousterian site of Sopena ~ rock-shelter (Asturias, Spain). *Quaternary Science Reviews* Volume 328, 15

We have included such citations. The Reviewer can find it in the introduction.

Pinto-Llona 2024 is reference 5

Yravedra 2011 is reference 6

Yravedra 2013 is reference 9

Sanchis 2019 is reference 14

Reviewer #4 (Remarks to the Author):

Many thanks.

3. Citations

- Bergström, Anders, Laurent Frantz, Ryan Schmidt, Erik Ersmark, Ophelie Lebrasseur, Linus Girdland-Flink, Audrey T. Lin, et al. 2020. "Origins and Genetic Legacy of Prehistoric Dogs." *Science* 370 (6516): 557–64.
- Bergström, Anders, David W. G. Stanton, Ulrike H. Taron, Laurent Frantz, Mikkel-Holger S. Sinding, Erik Ersmark, Saskia Pfrengle, et al. 2022. "Grey Wolf Genomic History Reveals a Dual Ancestry of Dogs." *Nature* 607 (7918): 313–20.
- Boschin, Francesco, Federico Bernardini, Elena Pilli, Stefania Vai, Clément Zanolli, Antonio Tagliacozzo, Rosario Fico, et al. 2020. "The First Evidence for Late Pleistocene Dogs in Italy." *Scientific Reports* 10 (1): 13313.
- Farrand, W. 2012. "Sedimentology of El Mirón Cave." In *El Mirón Cave, Cantabrian Spain: The Site and Its Holocene Archaeological Record*, edited by Lawrence Guy Straus and Manuel R. González Morales, 60–94. 9780826351487. University of New Mexico Press.
- Fu, Qiaomei, Cosimo Posth, Mateja Hajdinjak, Martin Petr, Swapan Mallick, Daniel Fernandes, Anja Furtwängler, et al. 2016. "The Genetic History of Ice Age Europe." *Nature* 534 (7606): 200–205.
- Geiling, J. M. 2020. "Human Ecodynamics in the Late Upper Pleistocene of Northern Spain: An Archeozoological Study of Ungulate Remains from the Lower Magdalenian and Other Periods" University of Cantabria.
- Hervella, Montserrat, Asier San-Juan-Nó, Aloña Aldasoro-Zabala, Koro Mariezkurrena, Jesús Altuna, and Concepción de-la-Rua. 2022. "The Domestic Dog That Lived ~17,000 Years Ago in the Lower Magdalenian of Erralla Site (Basque Country): A Radiometric and Genetic Analysis." *Journal of Archaeological Science: Reports* 46 (December): 103706.
- Kehl, M., D. Álvarez-Alonso, M. de Andrés-Herrero, P. Carral González, E. García, J. F. Jordá Pardo, M. Menéndez, et al. 2018. "Towards a Revised Stratigraphy for the Middle to Upper Palaeolithic Boundary at La Güelga (Narciandi, Asturias, Spain). Soil Micromorphology and New Radiocarbon Data." *Boletín Geológico Y Minero* 129 (1-2): 183–206.
- Loog, Liisa, Olaf Thalmann, Mikkel-Holger S. Sinding, Verena J. Schuenemann, Angela Perri, Mietje Germonpré, Herve Bocherens, et al. 2020. "Ancient DNA Suggests Modern Wolves Trace Their Origin to a Late Pleistocene Expansion from Beringia." *Molecular Ecology* 29 (9): 1596–1610.
- Marín-Arroyo, Ana B., Jeanne-Marie Geiling, Jennifer R. Jones, Manuel R. González Morales, Lawrence G. Straus, and Michael P. Richards. 2020. "The Middle to Upper Palaeolithic Transition at El Mirón Cave (Cantabria, Spain)." *Quaternary International: The Journal of the International Union for Quaternary Research* 544 (April): 23–31.
- Montes, R., J. Sanguino, P. Martín, A. J. Gómez, and C. Morcillo. 2005. "La Secuencia Estratigráfica de La Cueva de El Pendo (Escobedo de Camargo, Cantabria): Problemas Geoarqueológicos de Un Referente Cronocultural." *Geoarqueología Y Patrimonio En La Península Ibérica Y El Entorno Mediterráneo, ADEMA, Almazán (Soria)*, 127–38.
- Paijmans, Johanna L. A., Axel Barlow, Daniel W. Förster, Kirstin Henneberger, Matthias

- Meyer, Birgit Nickel, Doris Nagel, et al. 2018. "Historical Biogeography of the Leopard (*Panthera Pardus*) and Its Extinct Eurasian Populations." *BMC Evolutionary Biology* 18 (1): 1–12.
- Posth, Cosimo, He Yu, Ayshin Ghalichi, Hélène Rougier, Isabelle Crevecoeur, Yilei Huang, Harald Ringbauer, et al. 2023. "Palaeogenomics of Upper Palaeolithic to Neolithic European Hunter-Gatherers." *Nature* 615 (7950): 117–26.
- Sánchez-Romero, Laura, Alfonso Benito-Calvo, Ana B. Marín-Arroyo, Lucía Agudo-Pérez, Theodoros Karampaglidis, and Joseba Rios-Garaizar. 2020. "New Insights for Understanding Spatial Patterning and Formation Processes of the Neanderthal Occupation in the Amalda I Cave (Gipuzkoa, Spain)." *Scientific Reports* 10 (1): 8733.
- Sanchis, Alfred, Cristina Real, Víctor Sauqué, Carmen Núñez-Lahuerta, Natalia Égüez, Carmen Tormo, Manuel Pérez Ripoll, Yolanda Carrión Marco, Elsa Duarte, and Marco de la Rasilla. 2019. "Neanderthal and Carnivore Activities at Llonin Cave, Asturias, Northern Iberian Peninsula: Faunal Study of Mousterian Levels (MIS 3)." *Comptes Rendus. Palevol* 18 (1): 113–41.
- Sanguino, Juan, and Ramón Montes. 2001. *La cueva del "El Pendo" : actuaciones arqueológicas 1994-2000*. Consejería de Cultura, Educación y Deporte.
- Skoglund, Pontus, Erik Ersmark, Eleftheria Palkopoulou, and Love Dalén. 2015. "Ancient Wolf Genome Reveals an Early Divergence of Domestic Dog Ancestors and Admixture into High-Latitude Breeds." *Current Biology: CB* 25 (11): 1515–19.
- Straus, Lawrence G., and Manuel R. González Morales. 2018. "New Dates for the Solutrean and Magdalenian of Cantabrian Spain: El Miron and La Riera Caves." *Radiocarbon* 60 (3): 1013–16.
- Straus, Lawrence Guy, and Manuel R. González Morales. 2012. *El Mirón Cave, Cantabrian Spain: The Site and Its Holocene Archaeological Record*. University of New Mexico Press.
- Taron, Ulrike H., Johanna L. A. Paijmans, Axel Barlow, Michaela Preick, Arati Iyengar, Virgil Drăgușin, Ștefan Vasile, Adrian Marciszak, Martina Roblíčková, and Michael Hofreiter. 2021. "Ancient DNA from the Asiatic Wild Dog (*Cuon Alpinus*) from Europe." *Genes* 12 (2). <https://doi.org/10.3390/genes12020144>.
- Thalmann, O., B. Shapiro, P. Cui, V. J. Schuenemann, S. K. Sawyer, D. L. Greenfield, M. B. Germonpré, et al. 2013. "Complete Mitochondrial Genomes of Ancient Canids Suggest a European Origin of Domestic Dogs." *Science* 342 (6160): 871–74.
- Villalba-Mouco, Vanessa, Marieke S. van de Loosdrecht, Adam B. Rohrlach, Helen Fewlass, Sahra Talamo, He Yu, Franziska Aron, et al. 2023. "A 23,000-Year-Old Southern Iberian Individual Links Human Groups That Lived in Western Europe before and after the Last Glacial Maximum." *Nature Ecology & Evolution* 7 (4): 597–609.

Response to reviewers

We thank the reviewers for their constructive comments. We think that this manuscript has gained a lot during the editorial process. We also thank the editor for this.

Reviewer #1 (Remarks to the Author):

For my part, this new version of the manuscript is perfectly publishable. I consider it is a contribution of great value for research in the north of the Iberian Peninsula. And in this case, the new manuscript has greatly improved the issues that I raised, e.g. the poor contextualization of the site or other problems of archaeological nature. I can only congratulate the authors and I hope to see the paper published.

In any case, as I said on the previous occasion, my specialty is not DNA, so I consider that the review of the technical parts of the article should be assessed by specialists in the field.

I have noted minor issues related to the use of italics words in the bibliography. Please, review the bibliographic list in order to use italics when species are referenced, as in the case of *Cuon*, e.g.

Pérez Ripoll, M., Morales Pérez, J. V., Sanchis Serra, A., Aura Tortosa, J. E. & Montañana, I. S. Presence of the genus *Cuon* in upper Pleistocene and initial Holocene sites of the Iberian Peninsula: new remains identified in archaeological contexts of the Mediterranean region. *J. Archaeol. Sci.* 37, 437–450 (2010).

We have revisited the bibliography and we have corrected it.

Reviewer #2 (Remarks to the Author):

We have reviewed the initial submission of the manuscript titled "A sedimentary ancient DNA perspective on human and carnivore persistence through the Late Pleistocene in El Mirón Cave, Spain" by Gelabert et al., and we are generally very pleased with the revised version. The updated Figure 2, which illustrates the proportions of reads assigned to taxa,

the experiment addressing inhibition, the discussion of limitations, the enhanced clarity of the results, and the more cautious interpretation of findings, along with the extensive supplementary material, all significantly improve the manuscript. The article is now more engaging, and the narrative is better articulated.

However, we still have additional points that we believe the authors should address:

Major Points:

1. Line 118 – In the introduction, when El Mirón Cave is first mentioned, please include the region and country, as is done in the abstract. This mention should be followed by two or three sentences that contextualize the site and its relevance. The description provided in Supplementary Information 1 about the site generates more excitement to explore the results than the current introduction does.

We have added the location.

We have added the following sentences: “The site has a culture-stratigraphic sequence of levels dated by 101 radiocarbon determinations from >46,000 to c. 4000 cal BP (Straus and González Morales 2018; Straus and González Morales 2012). Remarkably, it exhibits excellent organic preservation. The faunal record has been studied and published and it was continuously occupied through the LGM, making it an ideal site to study the genomics of the humans and fauna through this period.” in Line 120.

2. Line 133 – The authors should provide more details about the design of their mtDNA probe set. It appears to be a new array not identical to the reference by Tejero et al., which is cited in the supplement for probe set design. Tejero et al.'s reference includes 54 mtDNA sequences, while this set contains 51. Although I found the list of species in Tejero et al.'s set, I could not locate the specific probe design strategy.

We have used the same kit as in Tejero et al 2024, with the only correction that some of the species were duplicated in the capture as an error of manufacturing. The product however includes 51 unique species, this explains the discrepancies between 54 and 51. The capture

is a commercial product by TWIST. We have added the following sentence “TWIST biosciences designed the capture as a custom product with probes of 80 bp covering the whole sequences of the mtDNA of the 51 species.” in line 240 of the SI.

We have added the new Supplementary Table 12 which provides the information about the sequences included in the capture.

3. Lines 357-358 – The authors state: "All these Fournol-ancestry individuals have haplogroup U2'3'4'7'8'9, restricted to Southern Europe and may represent an early population of humans with Fournol genetic ancestry in Western Europe."

We have rewritten this sentence (see question 4).

o The term "Fournol-ancestry" needs to be clearly defined, as its meaning is not immediately apparent to all readers.

We have added the following sentence: “The Fournol-ancestry refers to the Gravettian-like ancestry defined by an individual from Southern France that is thought to represent genetically the Iberian Solutrean genetic diversity (Posth et al. 2023)” in line 353.

o The sentence suggests that this haplogroup is restricted to Southern Europe, but Posth et al. (2016) shows haplogroup U2'3'4'7'8'9 throughout Europe, including as far east as Russia. Please clarify if there is a misunderstanding.

There is no individual with this haplogroup from Russia. The list of individuals with this haplogroup is:

Individual	Site	Haplogroup	Country
RIE002	La Riera	U2'3'4'7'8'9	Spain
LMA001	La Marche	U2'3'4'7'8'9	France
MAZ001	Maszycka	U2'3'4'7'8'9	Poland

MAZ003	Maszycka	U2'3'4'7'8'9	Poland
PIN004	Pincevent	U2'3'4'7'8'9	France
Rigney1		U2'3'4'7'8'9	France
BAL003.A0101	Spain_BalmaGuilanya	U2'3'4'7'8'9	Spain
I2158_v2	Italy_OrienteC	U2'3'4'7'8'9	Italy
UZZ5054	Italy_Uzzo_EM	U2'3'4'7'8'9	Italy
UZZ096	Italy_Uzzo_EM	U2'3'4'7'8'9	Italy
RIP001	Riparo Tagliente	U2'3'4'7'8'9	Italy
ST0001	San Teodoro	U2'3'4'7'8'9	Italy
MLZ	Malalmuerzo	U2'3'4'7'8'9	Spain

We have added this sentence in line 340: "So far the Haplogroup U2'3'4'7'8'9 has been identified in twelve four individuals. These individuals belong to the Gravettian, Solutrean or Magdalenian cultures from Poland to the Iberian Peninsula."

o The phrase "early population with Fournol genetic ancestry in Western Europe" is unclear. Are Southern and Western Europe considered mutually exclusive in this context?

We have replaced this sentence by: "All these individuals belong to the Gravettian, Solutrean or Magdalenian cultures from Poland to the Iberian Peninsula.. ", This follows the genetic classifications of Posth et al 2023.

4. Lines 363-364 – The authors state: "So far, the Haplogroup U2'3'4'7'8'9 has only been identified in four individuals (Malalmuerzo, La Riera, Rigney, and Paglicci108)."

o Posth et al. (2023) lists 12 individuals with this haplogroup, including a sample from Poland. This contradicts the claim that the haplogroup is restricted to Southern Europe. Please verify or clarify this statement.

We agree that the statement was not accurate. We have reframed to the following one: "currently identified in individuals with genetic ancestry linked to Gravettian, Solutrean or

Magdalenian genetic groups and restricted to individuals that lived from 27,000 to around 13,000 years ago.” in line 375.

5. Line 380 – The statement “The finding supports the archaeological evidence...” should be revised. The quantity of mtDNA sequences retrieved should not be used as a direct proxy for past species abundance at the site. Less human aDNA might reflect a lower spread of human DNA rather than a smaller population. It is misleading to make such a statement without knowing the exact source of aDNA for all animals.

We have changed to: “. Our findings do not contradict with archaeological evidence indicating relatively limited human activity and significant carnivore presence during the Middle to Upper Palaeolithic transition in the rear vestibule of El Mirón Cave ⁴⁸. The sedaDNA analyses identified 28 taxa (6 carnivores and 21 herbivores and humans).”

We no longer establish a relationship, we just mention that the results are not contradictory.

We agree with the reviewer and we discuss this topic in the discussion: “ Secondly, linking the amount of DNA discovered to specific activities or animal presence is challenging due to uncertainties regarding the tissue or organic material origins (i.e., faeces, saliva, urine, or other fluids). Our data proves the presence of a notable amount of DNA from species without remains. Further studies on DNA tissue origin can link animal activities and DNA presence. Third, we stretched the capacities to study ancient genomes due to the small amount of available mtDNA Pleistocene genomes of relevant species such as dholes or leopards. This limits our analyses and the inferences we can make. Finally, a potential limitation lies in the risk of DNA not contemporaneously deposited with the sediment. However, our rigorous validation methods and comprehensive analyses are designed to address and minimise these concerns effectively. We have evidenced the capacity to exclude the possible contaminant taxa and individually verify the identifications. The genetic similarity of our data to other Pleistocene sequences and the temporal dynamics strongly

support the fact that the DNA is contemporaneous with the stratigraphic levels and available dates. ”

Minor Points:

1. Line 74 – "Assess" is spelled incorrectly.

We have corrected it.

2. Line 119 – There is a comma after the parenthesis instead of a period.

We have corrected it.

3. Line 123 – I recommend replacing 'undocumented genetic affinities' with 'undocumented genetic phylogeny,' as 'affinities' seems too vague.

We have followed the reviewer's suggestion and it is rewritten (line 128).

4. Line 237 – The mention of assembling mtDNA for "21 Cervus etc..." could be confusing, as one might think the number refer to a number of individual instead of a number of sample. It would be clearer to state that this refers to the number of samples, not individuals.

We have added this sentence "These numbers indicate the individual samples for each species that had a coverage depth greater than 5X." in Line 226.

5. Lines 175-176 – The sentence "We did not recover endogenous human DNA from level 130, but traces of contamination (Supplementary Table 3, Supplementary Section 3)" should probably refer to Supplementary Table 9, where deamination rates are presented, as there are no human results in Supplementary Table 3.

We have corrected it, and have added table S9.

6. Figure 1 – The sample ID numbers are not very visible unless zoomed in. Consider making the circles slightly larger and using bold numbers for better visibility.

We updated the figure and increased the circle and font size while making the numbers bold. However, we can't increase the size any further as this would cause increased visual overlap between sampling locations.

7. Figure 2 – The color label for *Panthera pardus* is inconsistent between panel A and panel B.

We have corrected it in a new figure.

8. Supplementary Material 2, Line 215 – The phrase "We applied contamination control measures to mitigate the effect of contamination" lacks specificity. Clarify the specific measures taken.

We refer the reviewer to the next sentence: "All samples were processed at the Palaeogenomics laboratory of the University of Vienna. We applied contamination control measures to mitigate the effect of modern DNA contamination. The samples were prepared in dedicated clean room facilities. We included negative controls at each step of the wet lab pipeline to control for potential contamination of reagents."

9. Supplementary Table 1 – The column "Sequenced reads" appears twice in the table.

10. Supplementary Table 7 – The first row corresponding to sample "El Miron_26" appears to have incorrectly reported numbers for some taxa, as the numbers are identical for *Capra ibex*, *Rupicapra rupicapra*, *Ovis aries*, *Bison bonasus*, and *Bos taurus*. Please verify the accuracy of these numbers.

This has been solved. It was a bug in the code, we thank the reviewer for having pointed this out.

11. Supplementary Figure 4 – Reporting R^2 values, as done in Supplementary Figure 3, would help readers assess the correlation between archaeological level age, relative abundance, and inhibition.

We provide a new figure with these values.

Reviewer #3 (Remarks to the Author):

Respect this manuscript, I have seen and evaluated the authors' comments and the modifications they have made, and from my perspective I agree with the modifications they have made.

Thanks.

Reviewer #4 (Remarks to the Author):

Thanks.